# Marked synergy by vertical inhibition of EGFR signaling in NSCLC spheroids shows SOS1 is a therapeutic target in EGFR-mutated cancer

Patricia L Theard, Erin Sheffels, Nancy E Sealover, Amanda J Linke, David J Pratico, Robert L Kortum*

Department of Pharmacology and Molecular Therapeutics, Uniformed Services University of the Health Sciences, Bethesda, United States

**Abstract** Drug treatment of 3D cancer spheroids more accurately reflects in vivo therapeutic responses compared to adherent culture studies. In EGFR-mutated lung adenocarcinoma, EGFR-TKIs show enhanced efficacy in spheroid cultures. Simultaneous inhibition of multiple parallel RTKs further enhances EGFR-TKI effectiveness. We show that the common RTK signaling intermediate SOS1 was required for 3D spheroid growth of EGFR-mutated NSCLC cells. Using two distinct measures of pharmacologic synergy, we demonstrated that SOS1 inhibition strongly synergized with EGFR-TKI treatment only in 3D spheroid cultures. Combined EGFR- and SOS1-inhibition markedly inhibited Raf/MEK/ERK and PI3K/AKT signaling. Finally, broad assessment of the pharmacologic landscape of drug-drug interactions downstream of mutated EGFR revealed synergy when combining an EGFR-TKI with inhibitors of proximal signaling intermediates SOS1 and SHP2, but not inhibitors of downstream RAS effector pathways. These data indicate that vertical inhibition of proximal EGFR signaling should be pursued as a potential therapy to treat EGFR-mutated tumors.

*For correspondence:
robert.kortum@usuhs.edu

Competing interests: The authors declare that no competing interests exist.

## Introduction

Lung cancer is the leading cause of cancer-related death worldwide; adenocarcinomas are the most common subtype of lung cancer. Oncogenic driver mutations in the RTK/RAS pathway are found in over 75% of lung adenocarcinomas (*Cancer Genome Atlas Research Network, 2014*). Activating EGFR mutations occur in 10–30% of lung adenocarcinomas and are the major cause of lung cancer in never-smokers. In patients whose tumors harbor either an L858R mutation or an exon 19 deletion (85% of EGFR mutated tumors), first-generation EGFR-tyrosine kinase inhibitors (TKIs) erlotinib and gefitinib enhance progression-free survival (*Mok et al., 2009*; *Yang et al., 2017*; *Eberhard et al., 2005*). However, resistance to first generation EGFR-TKIs invariably occurs. In most cases, acquired resistance to first generation EGFR-TKIs occurs via either a secondary EGFR 'gatekeeper mutation' (T790M, 50–60% of cases) that renders the receptor insensitive to first generation EGFR-TKIs or oncogenic shift to alternative RTKs (15–30%). To treat patients with T790M-mutated resistant tumors, the third generation EGFR-TKI osimertinib, which selectively targets activating EGFR mutant proteins including T790M but spares wild-type EGFR, was developed (*Jänne et al., 2015*; *Cross et al., 2014*). However, despite further enhancing survival of patients with EGFR-mutant tumors, resistance again emerges.

Unlike first-generation EGFR-TKIs, mechanisms driving osimertinib resistance are more variable, including both EGFR-dependent (10–30%) and EGFR-independent mechanisms (*Mancini et al., 2018*; *Romaniello et al., 2018*; *La Monica et al., 2017*; *Eberlein et al., 2015*). The most common

**eLife digest** Lung cancer is the leading cause of cancer-related deaths worldwide. In non-smokers, this disease is usually caused by a mutation in a protein found on the surface of a cell, called EGFR. In healthy lung cells, these proteins trigger a chain of chemical signals that tell the cells to multiply. However, faulty forms of EFGR make the cells grow uncontrollably, leading to the formation of tumors.

Current treatments use EGFR inhibitors that block the activity of these proteins. But cancer cells often become resistant to these treatments by activating other types of growth proteins. One way to overcome this resistance has been by targeting the signaling pathways within individual tumors. But since those pathways differ between tumors, it has been challenging to find a single therapy that can treat all drug-resistant cancer cells.

Now, Theard et al. assessed the therapeutic effects of blocking a specific protein inside lung cells, called SOS1, which is involved in growth signaling in all tumor cells. Six different types of human lung cancer cells were used, all of which had faulty forms of EGFR, with three of the cell types showing drug resistance to current therapies. The cancer cells were either exposed to EGFR inhibitors only or to a combination of EGFR and SOS1 inhibitors. The most effective treatment was found to be through combinational therapy, with enhanced killing of drug-resistant cells.

Theard et al. further assessed the effect of combinational therapy using cells kept in two different ways. Cancer cells were either grown in a two-dimensional format, with cells forming a single cell layer, or in a three-dimensional format, where cells were multi-layered and grew on top of each other as self-aggregating spheroids. Combinational therapy treatment was only successful when the cells where grown in a three-dimensional format.

These findings highlight that future drug development studies should give consideration to the way cells are grown, as it can impact the results. They also provide a steppingstone towards tackling drug resistance in lung cancers that arise from EGFR mutations.

EGFR-independent resistance mechanisms involve reactivation of the RTK/RAS/effector pathway (*Eberlein et al., 2015*), often via enhanced signaling through parallel RTKs (*Mancini et al., 2018*; *Romaniello et al., 2018*; *La Monica et al., 2017*; *Shi et al., 2016*; *Park et al., 2016*; *Kim et al., 2019*; *Taniguchi et al., 2019*; *Jimbo et al., 2019*; *Namba et al., 2019*). Here, combining osimertinib with individual RTK inhibitors can both inhibit the development of resistance through the inhibited RTK and kill cancer cells with resistance driven by the specific RTK being inhibited. However, simultaneous inhibition of multiple RTKs with osimertinib may be required to eliminate oncogenic shift to alternative RTKs (*Romaniello et al., 2018*). Downstream of RAS, co-targeting intermediates of the RAF/MEK/ERK and PI3K/AKT pathways enhances of osimertinib effectiveness, however, signaling through the uninhibited effector pathway may drive resistance (*Tricker et al., 2015*; *Jacobsen et al., 2017*; *Ku et al., 2018*; *Ichihara et al., 2017*). Thus, it may be important for therapeutic combinations including osimertinib to stifle all downstream RTK/RAS signaling to be effective.

Recent studies suggest that pharmacologic assessments of targeted therapeutics should be performed under 3D culture conditions rather than in 2D adherent cultures (*Nunes et al., 2019*; *Langhans, 2018*). 3D spheroids show altered growth characteristics, changes in cell surface proteins, altered metabolism, changes in activation of signaling pathways or altered responses to targeted pathway inhibitors, and are more resistant to drug-induced apoptosis compared to 2D adherent cultures signaling (*Hao et al., 2019*; *Kim et al., 2011*; *Riedl et al., 2017*; *Jones et al., 2019*). These differences may be particularly relevant in *EGFR*-mutated NSCLC. *EGFR*-mutated cells show differential RTK expression and phosphorylation in 3D versus 2D conditions (*Ekert et al., 2014*). Further, EGFR-mutated cells respond more robustly to first-generation EGFR-TKIs in 3D cultures, and these responses more closely resemble responses seen in vivo (*Jacobi et al., 2017*). These data highlight the need for pharmacologic assessment of therapeutics designed to treat *EGFR*-mutated NSCLC under 3D culture conditions.

The ubiquitously expressed RasGEFs (guanine nucleotide exchange factors) SOS1 and SOS2 (son of sevenless 1 and 2) are common signaling intermediates of RTK-mediated RAS activation. Although not initially considered as drug targets because of the low oncogenic potential of SOS

(*Vigil et al., 2010*), there has been renewed interest in SOS proteins as therapeutic targets for cancer treatment. We and others have shown that SOS1 and SOS2 may be important therapeutic targets in KRAS-mutated cancer cells (*Jeng et al., 2012*; *Sheffels et al., 2018*; *Sheffels et al., 2019*), and a specific SOS1 inhibitor (BAY-293) has recently been identified (*Hillig et al., 2019*). Here, we investigate SOS1 and SOS2 as potential therapeutic targets in EGFR-mutated lung adenocarcinoma cells. Using two distinct measures of pharmacologic synergy, we demonstrate that SOS1 inhibition using BAY-293 synergizes with osimertinib only under 3D spheroid culture conditions, and in doing so add to the growing evidence that pharmacologic assessment of novel therapeutics designed to treat cancer must be performed under 3D culture conditions (*Ekert et al., 2014*; *Sheffels et al., 2018*; *Nunes et al., 2019*; *Janes et al., 2018*; *Jacobi et al., 2017*). By assessing the pharmacologic landscape of EGFR/RAS pathway inhibitors, we demonstrate that inhibition of proximal signaling is required to synergize with osimertinib, and that combined EGFR and SOS1 inhibition synergizes to inhibit RAS effector signaling in 3D culture. These findings have significant therapeutic implications for the development of combination therapies to treat EGFR-mutated lung adenocarcinoma.

## Results

### SOS1 deletion inhibits transformation in EGFR-mutated NSCLC cells

Previous studies showed that EGFR-mutated NSCLC cell lines show much more robust responsiveness to first-generation EGFR-TKIs in 3D culture (monoculture cancer cell line spheroids or monoculture or mixed culture organoids in ECM/Matrigel) compared to 2D adherent culture, and further that 3D conditions more readily mirror EGFR-TKI responses seen in vivo (*Jacobi et al., 2017*). To confirm these findings and extend them to third-generation EGFR-TKIs, we assessed dose-dependent survival of both first-generation EGFR-TKI sensitive (HCC827, exon 19 deletion [Δex19]) or resistant (NCI-H1975, L858R/T790M) NSCLC cell lines to either gefitinib or osimertinib treatment under both adherent (2D) or spheroid (3D) culture conditions (*Figure 1A*). HCC827 and H1975 cells were plated in either adherent or spheroid cultures, allowed to rest for 48 hr, and then treated with increasing doses of either the first-generation EGFR-TKI gefitinib or the third-generation EGFR-TKI osimertinib for 4 days. HCC827 cells showed responsiveness to both EGFR-TKIs under 2D and 3D culture conditions, however in both cases 3D spheroid cultures showed a > 1 log enhancement in EGFR-TKI efficacy and enhanced overall growth inhibition. While NCI-H1975 cells were not sensitive to gefitinib, osimertinib treatment of H1975 cells showed enhanced efficacy and increased overall growth inhibition in 3D spheroids over 2D adherent cultures.

SOS1 and SOS2 are ubiquitously expressed RasGEFs responsible for transmitting EGFR signaling to downstream effector pathways. To determine whether SOS1 or SOS2 were required for 2D anchorage-dependent proliferation or 3D spheroid growth in EGFR-mutated NSCLC cells, *SOS1* (*Figure 1—figure supplement 1* and *Munoz et al., 2016*) or *SOS2* (*31*) were deleted in pooled populations of HCC827 and H1975 cells to avoid clonal effects, and both proliferation and spheroid growth were assessed versus NT controls (*Figure 1B and C*). In adherent culture, neither *SOS1* nor *SOS2* deletion altered proliferation (*Figure 1B*). In contrast, *SOS1* deletion completely inhibited spheroid growth in both HCC827 and H1975 cells, indicating that SOS1 was required to maintain the transformed phenotype in both cell lines. To determine whether SOS1 was generally required for mutant EGFR-driven transformation, we further deleted *SOS1* or *SOS2* in both first-generation sensitive NCI-H3255 (L858R) and PC9 (Δex19) cells and in subcultures of these cell lines that had acquired T790M mutations after continuous EGFR-TKI treatment (PC9-TM [*de Bruin et al., 2014*] and H3255-TM [*Engelman et al., 2006*]). In all cases, *SOS1* deletion significantly diminished oncogenic transformation, whereas *SOS2* deletion had variable effects on transformation depending on the EGFR mutated cell line examined (*Figure 1D*). These data indicate that SOS1 is the major RasGEF responsible for oncogenesis downstream of mutated EGFR.

BAY-293 was recently described as a specific inhibitor for SOS1 (*Hillig et al., 2019*). To determine whether SOS1 inhibition was similarly more effective in 3D spheroids over 2D adherent culture, we assessed dose-dependent survival of H1975 cells after BAY-293 treatment under both 2D and 3D culture conditions (*Figure 1E*). Similar to what we observed after either EGFR-TKI treatment (*Figure 1A*) or *SOS1* deletion (*Figure 1C and D*), BAY-293 showed enhanced efficacy and increased overall growth inhibition in 3D spheroids over 2D adherent cultures. To confirm the specificity of

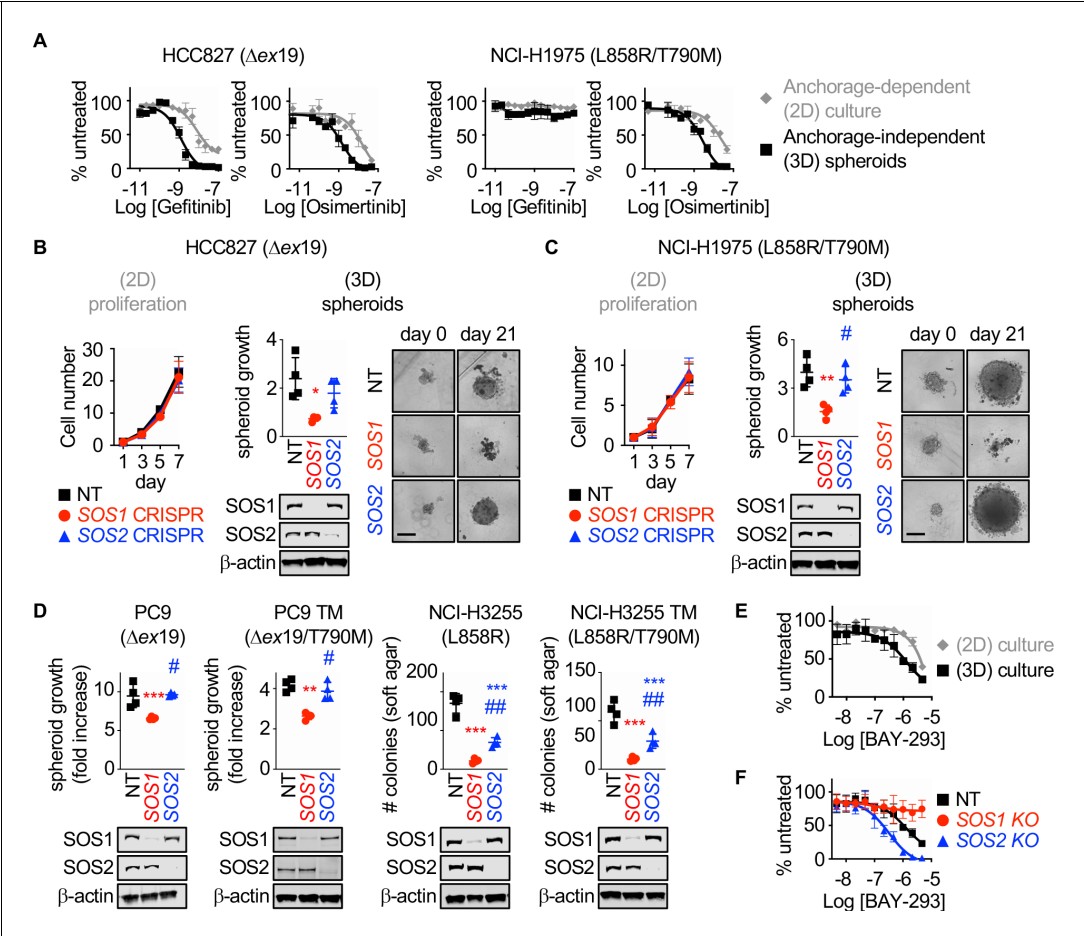

**Figure 1.** *SOS1* deletion inhibits anchorage-dependent (3D) transformation in EGFR-mutated NSCLC cell lines. (**A**) Dose-response curves of EGFR-mutated HCC827 (Δ*ex*19) (left) or NCI-H1975 (L858R/T790M) (right) cells treated with gefitinib or osimertinib under 2D anchorage-dependent (gray diamonds) or 3D spheroid (black squares) culture conditions. (**B-C**) 2D proliferation (left) or 3D spheroid growth (right) in pooled populations of (**B**) HCC827 or (**C**) NCI-H1975 cells where *SOS1* or *SOS2* has been deleted using CRISPR/Cas9 vs NT controls. 10x images of representative spheroids at day 0 and 21 are shown, scale bar = 250 mm. (**D**) 3D transformation in pooled populations of the indicated EGFR-mutated NSCLC cell lines where *SOS1* or *SOS2* has been deleted using CRISPR/Cas9 vs NT controls. (**E**) Dose-response curve cells of NCI-H1975 cells treated with the SOS1 inhibitor BAY-293 under 2D anchorage-dependent (gray diamonds) or 3D spheroid (black squares) culture conditions. Data are represented as cell # versus untreated for each individual cell line. (**F**) Dose-response curves of NCI-H1975 cells where *SOS1* (red circles) or *SOS2* (blue triangles) has been deleted using CRISPR/Cas9 vs NT controls (black squares) treated with BAY-293 under 3D spheroid culture conditions. For each condition, the untreated sample was set to 100%, and drug-treated samples were compared to untreated for each cell line. Dose-response curves and 2D proliferation are presented as mean +/- s.d. from a least three independent experiments. For transformation studies, data are from four independent experiments. Each individual experiment was performed using populations (not clones) of independently CRISPR'd cells. For each experiment, three technical replicates were assessed. Statistical significance was determined by ANOVA using Tukey's method for multiple comparisons. *p<0.05, **p<0.01, ***p<0.001 vs. NT cells. # p<0.05, ##p<0.01 vs. *SOS1* KO cells.

The online version of this article includes the following source data and figure supplement(s) for figure 1:

**Source data 1.** The SOS1 inhibitor BAY-293 is specific for SOS1 and is enhanced by*SOS2*deletion in EGFR (T790M) mutated NSCLC cell lines.
**Figure supplement 1.** Deletion of SOS1 using CRISPR/Cas9.
**Figure supplement 2.** The SOS1 inhibitor BAY-293 is specific for SOS1 and is enhanced by *SOS2* deletion in EGFR (T790M) mutated NSCLC cell lines.

BAY-293 for SOS1, we further treated 3D spheroid cultured H1975, PC9-TM, and H3255-TM cells where either *SOS1* or *SOS2* had been deleted versus NT controls with increasing doses of BAY-293 for four days, and assessed cell viability within the spheroids using Cell Titre Glo (*Figure 1F* and *Figure 1—figure supplement 2*). BAY-293 treatment did not inhibit survival of spheroids where *SOS1* had been deleted, indicating the specificity of BAY-293 for SOS1. Further, cells where *SOS2* had been deleted showed an approximately 1-log enhancement in BAY-293 efficacy and enhanced

overall growth inhibition compared to NT controls, indicating that SOS1 and SOS2 have over-lapping functions in supporting survival of spheroid cultured EGFR-mutated NSCLC cells. For these experiments, the untreated sample cell number at day four of treatment for each cell line (NT, *SOS1* KO, *SOS2* KO) was set to 100%, so differences in transformation (see *Figure 1B–D*) will not be appreciated. Further, for NCI-H1975 and NCI-H3255-TM cells, *SOS1* deletion does not show trans-formation differences after four days. Overall, these data suggest that EGFR-mutated NSCLC cells are more sensitive to either mutant EGFR or SOS1 inhibition in 3D spheroid culture compared to tra-ditional 2D adherent conditions.

## SOS1 inhibition synergizes with EGFR-TKIs to inhibit cell survival under anchorage independent (3D) culture conditions

Previous studies reported that combining osimertinib with an alternative RTK inhibitor may inhibit or treat the development of resistance driven by that specific RTK (*Mancini et al., 2018*; *Romaniello et al., 2018*; *La Monica et al., 2017*), whereas simultaneous inhibition of multiple paral-lel RTKs with osimertinib may be required to effectively potentiate osimertinib action (*Romaniello et al., 2018*). Further, while many studies show enhanced drug activity in combination therapies versus osimertinib treatment alone, they do not assess whether the effects of the two-drug combinations are truly synergistic; synergistic interactions between therapeutics allow for maximiza-tion of the therapeutic effect while minimizing adverse events and may be required for effective therapeutic combinations with targeted agents (*Roell et al., 2017*).

SOS1 is a common downstream mediator of RTK signaling. We hypothesized that SOS1 could be an effective drug target to synergize with EGFR-TKI inhibition to treat EGFR-mutated lung adenocar-cinoma. To directly assess synergy between osimertinib and SOS1 inhibition, we use two distinct methods based on the most widely established reference models of drug additivity. The first method, isobologram analysis, assesses changes in the dose-response curves for mixtures of two drugs compared to sham mixtures of each individual drug with itself. The second method, Bliss inde-pendence analysis, assesses whether a mixture of two individual drug doses has a greater effect than would be expected if the two drugs acted independently. We will first describe and then use each method in turn to determine the whether SOS1 inhibition using BAY-293 could synergize with the EGFR-TKI osimertinib in *EGFR*-mutated lung adenocarcinoma cells.

Isobologram analysis is a dose-effect analysis based on the principle of Loewe additivity, which states that a drug mixed with itself, and by extension a mixture of two or more similar drugs, will show additive effects. For two drugs (Drug A and Drug B) that have parallel dose-response curves so that a constant potency ratio is maintained at all doses of A and B (*Figure 2A*), treatment using any dose-equivalent (DEQ) mixture of Drugs A and B will show a similar effect to treatment with either Drug A or Drug B alone if the effects of the two drugs are additive. In contrast, if the two drugs show synergism, then the effect seen by treatment with DEQ mixtures of A and B will be greater than the effect for either drug alone. By generating dose-response curves for different DEQ mixtures of Drugs A and B (*Figure 2B*), one can compare the $EC_{50}$ of each DEQ mixture to the $EC_{50}$ of Drug A or Drug B alone on an isobologram plot (*Figure 2C*). The $EC_{50}$ of each individual drug is plotted as the x- or y-intercept, and the calculated contribution of each drug to the overall $EC_{50}$ for each DEQ mix is plotted as a single point ($EC_{50,A}$, $EC_{50,B}$) on the graph. If the $EC_{50}$ values for each DEQ mix fall along the straight line (isobole) that connects the individual drug $EC_{50}$ values, then the drug-drug interaction is additive. In contrast, points that fall above or below the isobole indicate antago-nism or synergy. The extent to which two drugs interact can be further quantified from the $EC_{50}$ data as a combination index (CI) (*Figure 2D*). A CI between 0.8 and 1.2 indicates the two drugs have additive effects when combined, a CI <0.8 indicates synergy, and a CI >1.2 indicates antagonism.

To assess drug-drug synergy between osimertinib and BAY-293 via isobologram analysis, NCI-H1975 cells were cultured under 2D adherent or 3D spheroid conditions for 48 hr, and were treated with varying DEQ combinations of osimertinib:BAY-293 (see *Figure 2B*) for four days. Cell viability data was assessed using CellTiter-Glo and $EC_{50}$ values from each DEQ mixture were used to gener-ate isobologram plots and calculate combination indices (*Figure 2E*). When cells were cultured under 2D conditions, osimertinib and BAY-293 showed additive effects, as DEQ $EC_{50}$ values fell on the isobole and CI values were between 0.8 and 1.2. In contrast, when cells were cultured as 3D

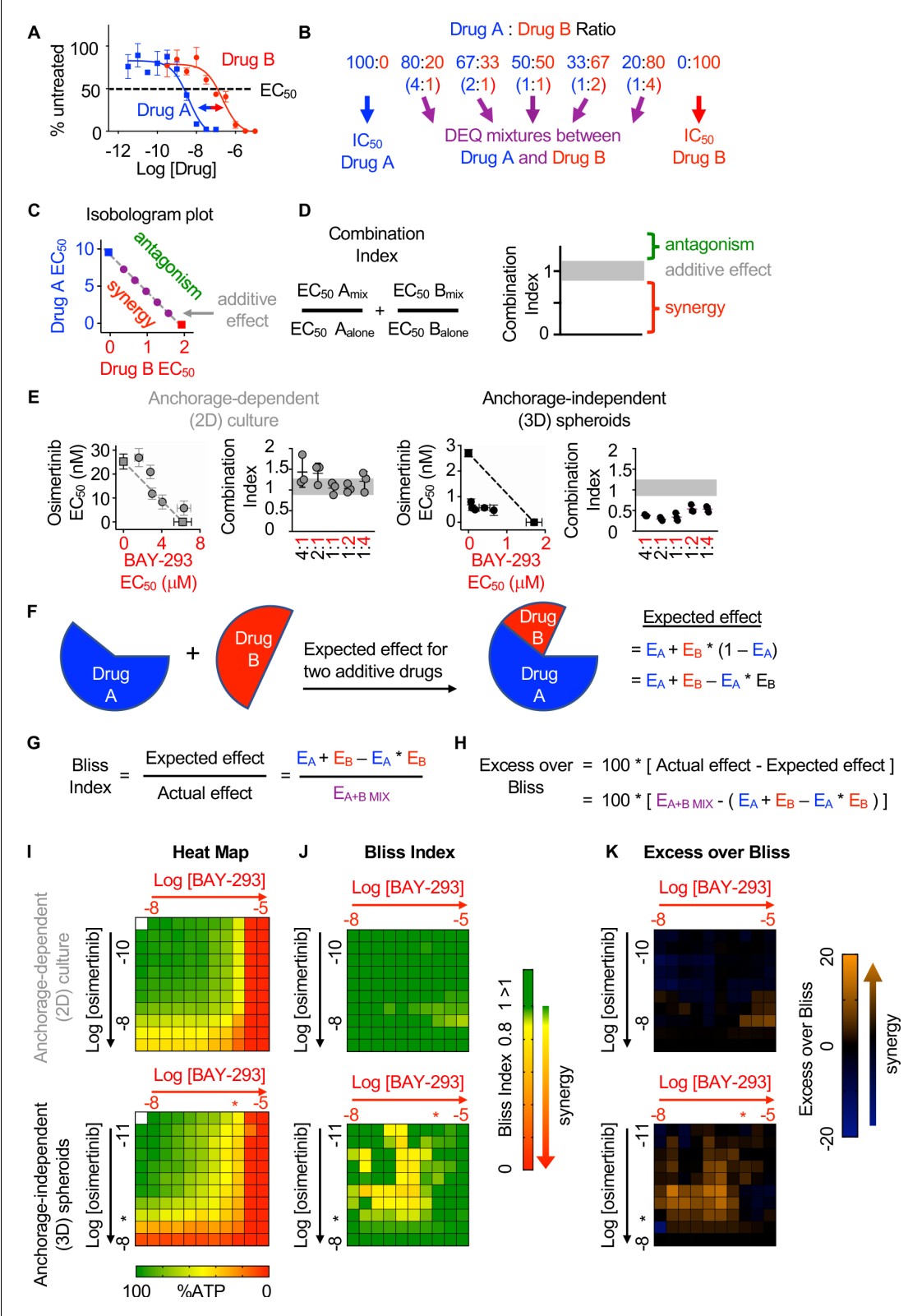

**Figure 2.** SOS1 inhibition synergizes with the EGFR-TKI inhibitor osimertinib to inhibit cell survival under anchorage-independent (3D) culture conditions. (A-D) Isobologram analysis examines drug-drug synergy by comparing dose equivalent (DEQ) mixtures of two drugs based on their $EC_{50}$ values to treatment with either drug alone (A and B). From the dose-response curves of the DEQ mixtures, plotting the fractional $EC_{50}$ for each drug in the combination (purple) relative to the individual drug $EC_{50}$ values (blue, red) on an isobologram plot (C) and calculation of the combination index (CI, *Figure 2 continued on next page*

*Figure 2 continued*

D and E) allows assessment of drug-drug synergy. Additive effects occur on the dashed lines of the isobologram plot and have a CI 0.8–1.2 (gray box), whereas synergistic interactions fall below the dashed lines and have a CI <0.8. (E) Isobologram plots and CI from dose-equivalent treatments of H1975 EGFR-mutated NSCLC cells treated with DEQ combinations of osimertinib and BAY-293. Isobologram and CI data are presented as mean +/- s.d. from three independent experiments. (F) Bliss additivity evaluates whether the overall effect of an individual drug combination ($E_{A+B\ mix}$) is greater than should be expected for two drugs with independent effects on the overall population ($E_A + E_B – E_A * E_B$). (G) The Bliss Index compares the ratio of the expected effect to the actual effect. Synergistic interactions have a Bliss Index < 0.85. (H) Excess over Bliss evaluates the magnitude of the difference between the actual and expected effects. Increasingly synergistic interactions show an excess over Bliss Index > 0. (I) Heat map of H1975 cells treated with the indicated doses of osimertinib and/or BAY-293 grown in either 2D (adherent) culture conditions or as 3D spheroids. Green indicates more cells, red indicates fewer cells. $EC_{50}$ values for each individual drug are indicated by an *. (J) Heat map of Bliss Index assessing drug-drug synergy between osimertinib and BAY-293 at each dose combination from D. (K) Heat map of excess over Bliss assessing drug-drug synergy between osimertinib and BAY-293 at each dose combination from D. Bliss Index and excess-over Bliss are presented as the mean from three independent experiments. For each experiment, three technical replicates were assessed.

The online version of this article includes the following source data for figure 2:

**Source data 1.** SOS1 inhibition synergizes with the EGFR-TKI inhibitor osimertinib to inhibit cell survival under anchorage-independent (3D) culture conditions.

spheroids, osimertinib and BAY-293 showed significant synergy, as DEQ $EC_{50}$ values were well below the isobole and CI <0.8.

Bliss independence analysis is an effect-based analysis based on the principle of Bliss additivity, which assumes that two drugs will act independently of each other so that their combined effect can be assessed by assessing the effect of each drug sequentially (*Figure 2F*). Unlike isobologram analysis, this method does not require that two drugs being assessed have parallel dose-response curves and can be calculated based as few as three drug treatments, the effect each drug has on its own on the cell population, and the effect of combining the two drug treatments together. By representing the effect of each drug treatment as a probabilistic outcome between 0 (no effect) and 1 (100% effect), we can compare the observed effect of the drug-drug combination to the expected effect if each drug acted independently (*Figure 2E*). The ratio of the expected effect to the observed effect is the Bliss Index (BI), where a BI <1 indicates synergy (*Figure 2G*). Alternatively, the magnitude of the difference between the observed and expected result can be reported as the excess over Bliss (*Figure 2H*). While excess over Bliss is the most widely reported synergy metric, the Bliss Index can be directly compared with the combination index in isobologram experiments and should be used when both synergy methods are used to assess a given drug-drug interaction.

To assess drug-drug synergy between osimertinib and BAY-293 via Bliss Independence analysis, NCI-H1975 cells were cultured under 2D adherent or 3D spheroid conditions for 48 hr and were treated with increasing doses of BAY-293, osimertinib, or combinations of the two drugs over a 3-log scale for four days. Cell viability was determined using CellTiter-Glo and overall viability (*Figure 2I*), Bliss index (*Figure 2J*), and excess over Bliss (*Figure 2K*) were represented as heat-maps. Similar to what we observed for isobologram analysis, osimertinib and BAY-293 did not show significant synergy in cells cultured under 2D adherent conditions. In contrast, we observed significant synergy between osimertinib and BAY-293, mostly at dose combinations of osimertinib and BAY-293 falling just below the individual drug $EC_{50}$ values. Overall, the data presented in *Figure 2* indicate that osimertinib and BAY-293 show significant drug-drug synergy in EGFR-mutated H1975 cells, but only in 3D spheroid culture conditions.

To determine whether the SOS1 inhibitor BAY-293 could generally synergize with EGFR-TKIs in EGFR-mutated lung adenocarcinoma cells, we extended our assessment of drug-drug synergy to isobologram analysis (*Figure 3*) and Bliss independence analysis (*Figure 4*) in six different EGFR-mutated lung adenocarcinoma cell lines. In cells that were sensitive to first-generation EGFR-TKIs (HCC827, PC9, H3255; T790 wild-type), we assess drug-drug synergy between BAY-293 and either a first-generation (gefitinib) or third-generation (osimertinib) EGFR-TKI. In cells that were resistant to first-generation EGFR-TKIs (H1975; PC9-TM, H3255-TM; T790M) we limited our assessment to synergy between BAY-293 and osimertinib. To first determine the individual $EC_{50}$ values for gefitinib, osimertinib, and BAY-293 in each cell line, cells were cultured as 3D spheroids for 48–72 hr, and then treated with increasing doses of drug for four days followed by assessment of cell viability by CellTiter-Glo (*Figure 3—figure supplement 1*). In five of six cell lines, the individual dose-response

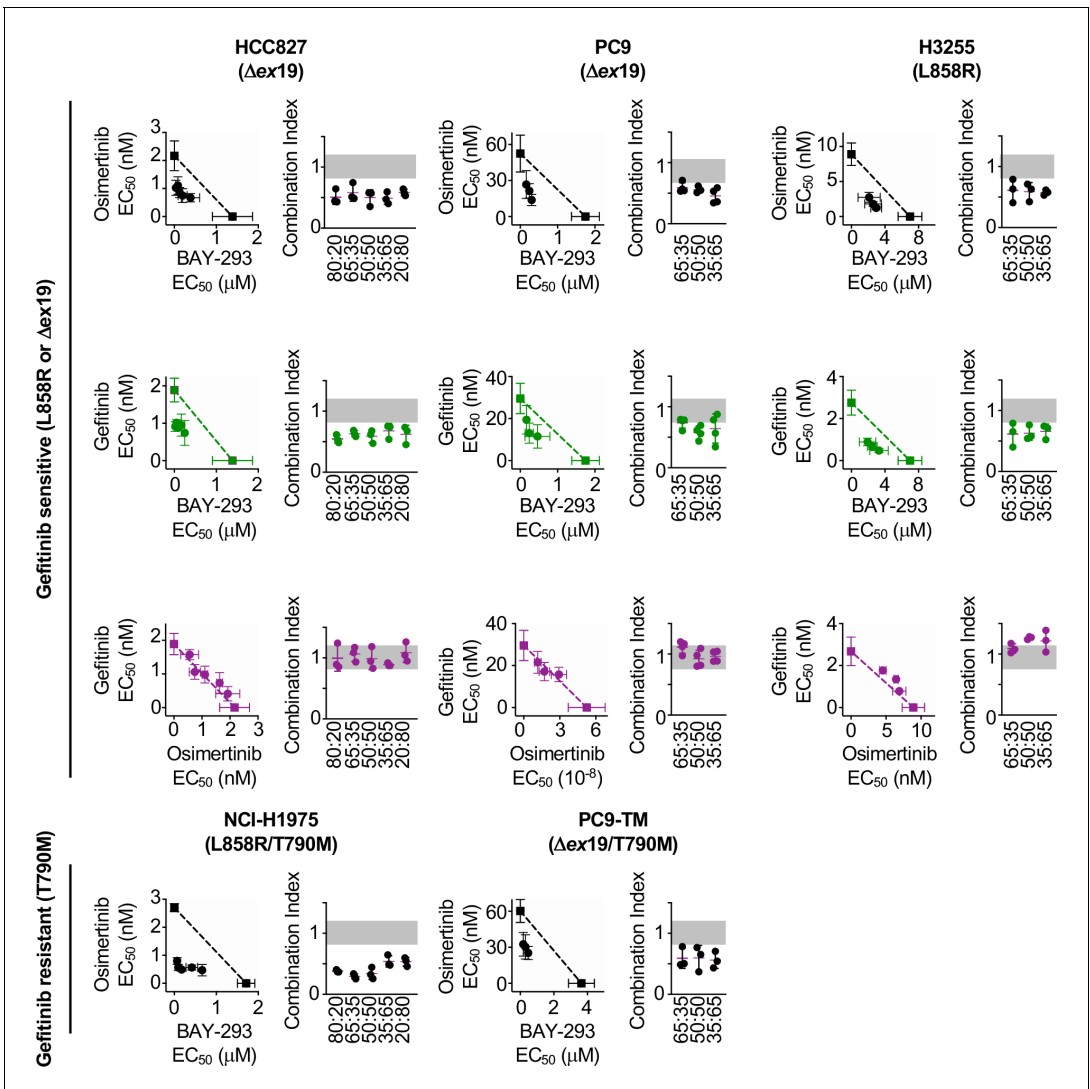

**Figure 3.** Isobologram analysis showing that SOS1 inhibition synergizes with EGFR-TKI treatment to inhibit survival in multiple EGFR-mutated NSCLC cell lines. Isobologram analysis and Combination Index (CI) from dose-equivalent treatments of the indicated EGFR-mutated gefitinib-sensitive (L858R or Δex19, top) or gefitinib-resistant (T790M, bottom) NSCLC cell lines with combinations of gefitinib, osimertinib, and BAY-293. Additive effects occur on the dashed lines of the isobologram plot and have a CI 0.8–1.2 (gray box), whereas synergistic interactions fall below the dashed lines and have a CI <0.8. Data are presented as mean +/- s.d. from three independent experiments. For each experiment, three technical replicates were assessed.
The online version of this article includes the following source data and figure supplement(s) for figure 3:

**Source data 1.** EGFR mutated NSCLC cell lines are responsive to osimertinib, BAY-293, and gefitinib in 3D spheroid cultures.
**Figure supplement 1.** EGFR mutated NSCLC cell lines are responsive to osimertinib, BAY-293, and gefitinib in 3D spheroid cultures.

curves for BAY-293, osimertinib, and gefitinib (where appropriate) showed similar maximal effects and Hill coefficients, and were thus appropriate for linear isobologram analysis for each two-drug combination of BAY-293, osimertinib, and gefitinib (*Tallarida, 2011*). In contrast, H3255-TM cells were only moderately sensitive to osimertinib, showing at most a 50% reduction in viability at high doses. Therefore, we limited our assessment of drug-drug synergy in H3255-TM cells to Bliss independence analysis. Further, to simplify our assessment of Bliss independence across multiple drugs and cell lines, we limited our drug treatments to 1:2, 1:1, and 2:1 mixtures of each drug combination based on dose equivalence (see *Figure 4A*).

For each first-generation EGFR-TKI sensitive cell line (HCC827, PC9, H3255), gefitinib and osimertinib did not show any synergy with each other by either isobologram analysis (*Figure 3*) or Bliss Independence analysis (*Figure 4*), instead showing additive effects (CI and BI ~1) as would be

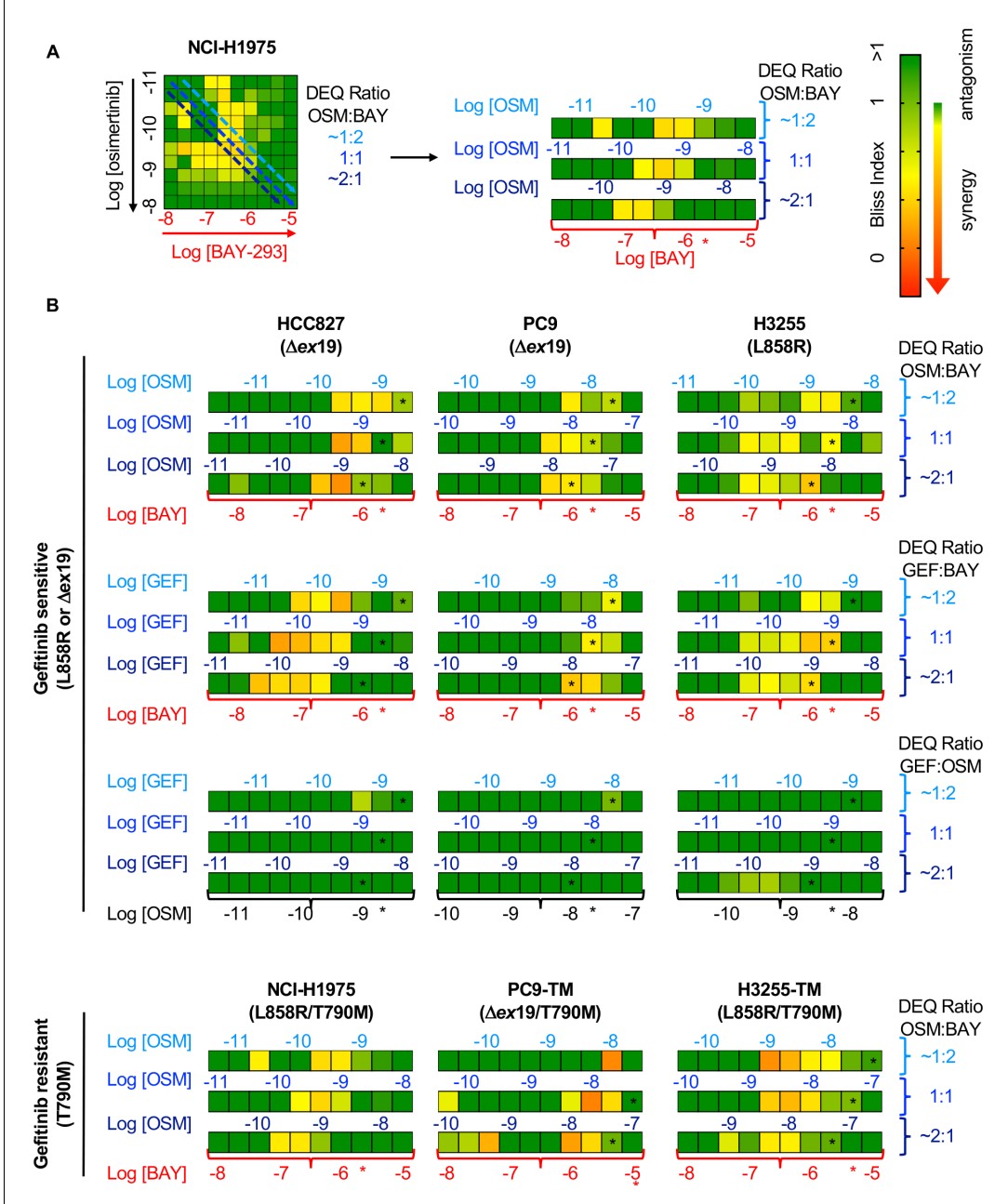

**Figure 4.** Bliss Independence analysis showing that SOS1 inhibition synergizes with EGFR-TKI treatment to inhibit survival in multiple EGFR-mutated NSCLC cell lines. (**A**) Bliss Index heatmap from 3D spheroid cultured NCI-H1975 cells *Figure 2A* (left) and horizontal projections of Bliss Indices of drug treatments at 2:1, 1:1, and 1:2 ratios of osimertinib:BAY-293 based on dose equivalencies (right). Increasingly synergistic interactions (Bliss index <0.85) are indicated by the corresponding heat map. The concentration of BAY-293 (held constant, bottom) and of osimertinib (above each horizontal projection) are given. The IC$_{50}$ for each individual drug are shown (*). (**B**) Bliss Index heatmaps based on A for the indicated gefitinib-sensitive and gefitinib-resistant cell lines at 2:1, 1:1, and 1:2 ratios of osimertinib, gefitinib, and BAY-293 based on dose equivalencies. Data for NCI-H1975 cells are the same as in A. Data are presented as the mean from three independent experiments. For each experiment, three technical replicates were assessed. The online version of this article includes the following source data for figure 4:

**Source data 1.** Bliss Independence analysis showing that SOS1 inhibition synergizes with EGFR-TKI treatment to inhibit survival in multiple EGFR-mutated NSCLC cell lines.

expected for two drugs with the same molecular target. In contrast, BAY-293 showed significant synergy with gefitinib and osimertinib by both isobologram analysis (*Figure 3*) and Bliss Independence analysis (*Figure 4*), suggesting that SOS1 inhibition can act as a secondary treatment for all EGFR-TKIs. Further, in all three T790M mutated cell lines (H1975, PC9-TM, H3255-TM), BAY-293 again showed synergy with osimertinib. These data suggest that combined SOS1 and EGFR inhibition is a robust therapeutic combination that synergize to inhibit EGFR-mutated lung adenocarcinoma cell growth.

## Synergy between BAY-293 and osimertinib is independent of SOS2

We showed that *SOS2* deletion sensitized NCI-H1975 cells to the SOS1 inhibitor BAY-293 (*Figure 1F*). We wanted to determine whether the synergy we observed between EGFR- and SOS1-inhibition (*Figures 3* and *4*) was enhanced by *SOS2* deletion in EGFR-mutated NSCLC cell lines. To examine whether SOS2 deletion alters the synergy between osimertinib and BAY-293 in EGFR (T790M) mutated cells, SOS2 was deleted in H1975, PC9-TM, and H3255-TM cells. For H1975 and PC9-TM cells, *SOS2* KO cells vs NT controls were cultured under 3D spheroid conditions for 48–72 hr, and were then treated with varying DEQ combinations of osimertinib:BAY-293 for 4 days. Cell viability data was assessed using CellTiter-Glo and $EC_{50}$ values from each DEQ mixture were used to generate Isobologram plots and calculate confidence intervals (*Figure 5A and B*). For both cell lines, *SOS2* deletion sensitized cells to BAY-293, decreasing $EC_{50}$ by 5–10-fold compared to NT controls without altering the $EC_{50}$ to osimertinib treatment alone. However, unlike what we observed in the NT control cells, osimertinib and BAY-293 showed only mild synergy in EGFR-mutated cells where *SOS2* was deleted as assessed by the distance of the interaction points to the isobole and the increased combination index vs. NT controls. Further, when we overlaid the NT and *SOS2* KO isobologram plots at two different scales of BAY-293, the drug combination data points were overlapping between NT and *SOS2* KO cells, suggesting that *SOS2* deletion did not enhance synergy between osimertinib and BAY-293.

Since H3255-TM cells are not appropriate for linear isobologram analysis between BAY-293 and osimertinib, we instead performed Bliss independence analysis to assess potential synergy between osimertinib and BAY-293 in the presence or absence of SOS2. H3255-TM cells where *SOS2* had been deleted vs NT controls were cultured under 3D spheroid conditions for 48–72 hr, and were then treated with increasing doses of osimertinib alone, BAY-293 alone, or mixtures of each drug dose at 1:2, 1:1, and 2:1 mixtures of osimertinib and BAY-293 based on dose equivalence for four days. Cell viability data was assessed using CellTiter-Glo, and the Bliss index was calculated for each drug mixture as shown in *Figure 2C* and *Figure 4*. As was the case in H1975 and PC9-TM cells, while the *SOS2* deletion sensitized H3255-TM cells to BAY-293 we observed less overall synergy between osimertinib and BAY-293 H3255-TM cells where we had deleted *SOS2* vs NT controls. These data suggest that although osimertinib and BAY-293 synergize to limit viability of EGFR-mutated lung adenocarcinoma cells, the synergy between osimertinib and BAY-293 is independent of SOS2.

## BAY-293 and osimertinib synergize to inhibit RAS effector signaling

Mutated EGFR signals through downstream RAF/MEK/ERK and PI3K/AKT effector pathways to promote proliferation, transformation, and survival. Since *SOS2* deletion did not further enhance synergy between BAY-293 and osimertinib, we hypothesized that SOS1 inhibition specifically enhanced EGFR-TKI-dependent inhibition of downstream signaling in 3D culture. To perform signaling experiments on 3D cultured spheroids, cells were seeded in 24-well micropatterned low-attachment culture plates (Aggrewell, StemCell) containing ~1200 individual spheroids per condition. To determine the extent to which SOS1 inhibition and/or *SOS2* deletion altered osimertinib-dependent inhibition of downstream effector signaling in 3D culture, H1975 or PC9-TM cells where *SOS2* was deleted vs. NT controls were cultured as spheroids for 48–72 hr and then treated with increasing doses of osimertinib +/- BAY-293 prior to spheroid collection, lysis, and western blotting for phosphorylated ERK and AKT (*Figure 6*). In both NT and *SOS2* knockout cells, BAY-293 reduced the dose of osimertinib required to inhibit both ERK and AKT phosphorylation (*Figure 6*). For Raf/MEK/ERK signaling, Bliss Independence analysis of pERK quantitation revealed that either SOS1 inhibition or *SOS2* deletion independently synergized with osimertinib to inhibit Raf/MEK/ERK signaling, and the combination of inhibiting SOS1/2 signaling further enhanced this synergy. In contrast, for PI3K/AKT signaling

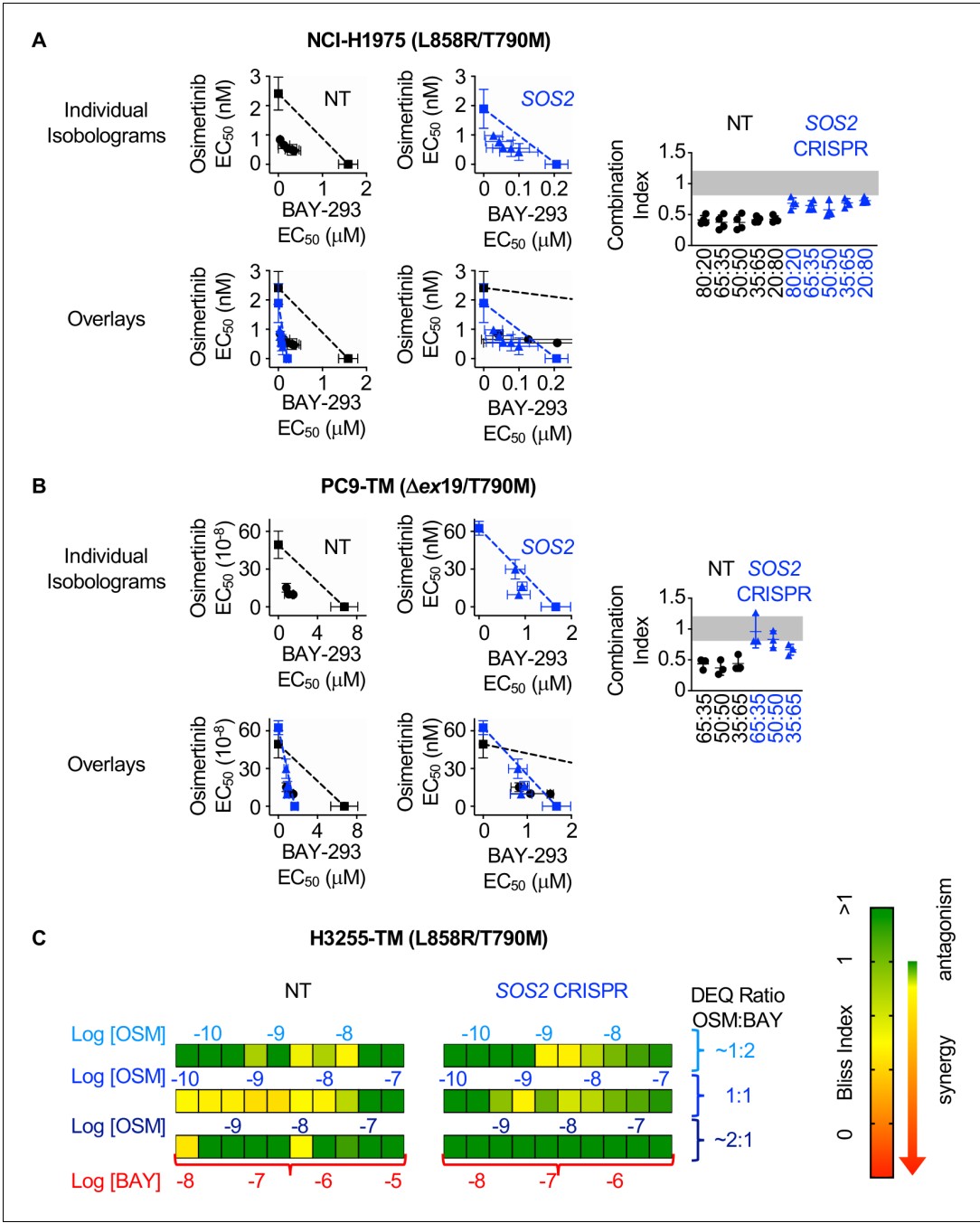

**Figure 5.** *SOS2* deletion does not enhance the synergistic interaction between SOS1 inhibition and EGFR-TKI treatment. (A-B) Isobologram analysis (left) and Combination Index (right) from dose-equivalent treatments of osimertinib and BAY-293 in H1975 (A) or PC9-TM (B) cells where *SOS2* has been deleted (blue) versus NT controls (black). Overlay plots on two different BAY-293 dosing scales are shown below the individual isobologram plots. Additive effects occur on the dashed lines of the isobologram plot and have a CI 0.8–1.2 (gray box), whereas synergistic interactions fall below the dashed lines and have a CI <0.8. (C) Bliss Index heatmaps for H3255-TM cells where *SOS2* has been deleted versus NT controls treated at at 1:2, 1:1, and 2:1 ratios of osimertinib and BAY-293 based on dose equivalencies. Data are presented as mean +/- s.d. from three independent experiments. For each experiment, three technical replicates were assessed.

The online version of this article includes the following source data for figure 5:

**Source data 1.** *SOS2* deletion does not enhance the synergistic interaction between SOS1 inhibition and EGFR-TKI treatment.

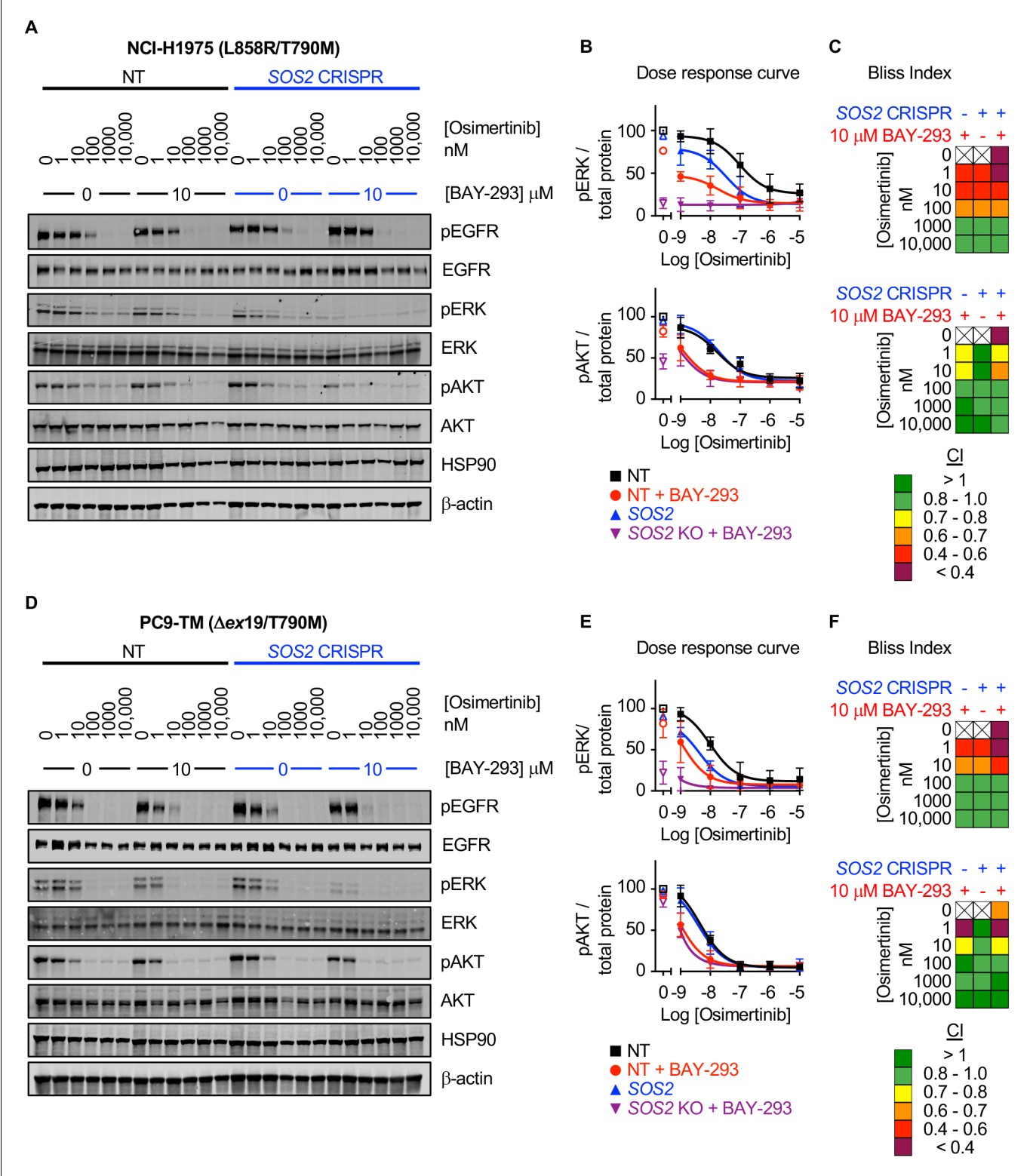

**Figure 6.** SOS1 inhibition synergizes with mutant EGFR inhibition to inhibit downstream effector signaling. Western blots (**A, D**), pERK and pAKT quantitation (**B, E**), and Bliss Indices (**C, F**) of WCLs of NCI-H1975 cells (**A-C**, top) or PC9-TM cells (**D-F**, bottom) cultured under 3D spheroid conditions for 48 hr and then treated with the indicated concentrations of the EGFR-TKI osimertinib and/or the SOS1 inhibitor BAY-293 for 6 hr. Western blots are for pEGFR, EGFR, pAKT, AKT, pERK1/2, ERK1/2, HSP90, and β-actin. pERK and pAKT quantifications were calculated using a weighted average of total protein western blots. Combination Indices are based on pERK/Total protein and pAKT/Total protein quantitations. Increasingly synergistic

*Figure 6 continued on next page*

Figure 6 continued

combinations are indicated in yellow, orange, red, or purple. Phosphoprotein quantitations are presented as mean +/- s.d. from three independent experiments. Bliss indices are presented as mean from three independent experiments. For each experiment, three technical replicates were assessed. The online version of this article includes the following source data for figure 6:

**Source data 1.** SOS1 inhibition synergizes with mutant EGFR inhibition to inhibit downstream effector signaling.

*SOS2* deletion did not enhance the synergy between osimertinib and BAY-293. While either osimertinib treatment or *SOS2* deletion independently synergized with BAY-293 to inhibit AKT phosphorylation, *SOS2* deletion did not further enhance the ability osimertinib to inhibit PI3K/AKT signaling in the presence or absence of BAY-293. These data strongly suggest that vertical inhibition of EGFR and SOS1 limits call viability by inhibiting activation of both RAF/MEK/ERK and PI3K/AKT effector pathways.

## Assessment of inhibitor landscape in EGFR-mutated cells lines shows synergy upon inhibition of upstream pathway effectors

Since the most common EGFR-independent resistance mechanisms involve reactivation of RTK/RAS/ effector pathways (*Mancini et al., 2018*; *Romaniello et al., 2018*; *La Monica et al., 2017*; *Eberlein et al., 2015*), we wanted to assess whether inhibition of different proteins within the EGFR/ RAS signaling pathway could synergize to inhibit 3D survival of EGFR (T790M) mutated cancer cells. To determine drug-drug synergies after inhibition of EGFR-RAS pathway signaling at different levels, we assessed synergy between osimertinib, inhibitors of EGFR signaling intermediates upstream of RAS (BAY-293 for SOS1 and RMC-4450 for SHP2), and inhibitors of the Raf/MEK/ERK (trametinib) and PI3K/AKT (buparlisib) pathways (*Figure 7A*). H1975 and PC9-TM cells were treated with each individual inhibitor or 1:1 DEQ mixtures of every drug-drug combination, and the combination index was calculated to assess drug-drug synergy. Since H3255-TM cells are not suitable for isobologram analysis, these cells were treated with full-dose mixtures based on dose equivalence and the Bliss Index was calculated for each drug-drug combination (*Figure 7B*). Intriguingly, all three cell lines showed drug-drug synergy with any combination of EGFR, SOS1, and SHP2 inhibition. In contrast, inhibition of downstream Raf/MEK/ERK or PI3K/AKT pathways failed to consistently synergize with either osimertinib or any other inhibitor (*Figure 7B*, top). These data support the premise that combined vertical inhibition of proximal EGFR signaling may constitute an effective strategy to treat EGFR-mutated lung adenocarcinomas.

SHP2 is important for the stabilization of the GRB2:SOS1/2 complexes on EGFR (*Dance et al., 2008*), and the mechanism of allosteric SHP2 inhibitors depends on SOS1 (*Nichols et al., 2018*), although the contribution of SOS2 to SHP2 inhibitors was not assessed. To determine whether *SOS2* deletion altered the spectrum of drug-drug synergies in EGFR-mutated cells, parallel studies were performed in EGFR-mutated cells where *SOS2* was deleted (*Figure 7B*, bottom). Unlike what we observed for synergy between EGFR- and SOS1 inhibition, synergy between SOS1 and SHP2 inhibition was enhanced by *SOS2* deletion. These data suggest that SOS2 plays a role in SHP2-dependent signaling. SOS1 inhibition also synergized with MEK inhibition in *SOS2* KO cells. Given the strong synergy between SOS1 inhibition and *SOS2* deletion in inhibiting Raf/MEK/ERK signaling (*Figure 6*), these data suggest that deep inhibition of MEK signaling is sufficient to inhibit survival in EGFR-mutated cells.

To further evaluate synergy between inhibitors of proximal EGFR signaling, we examined combinations of EGFR- SOS1- and SHP2 inhibition both by expanded evaluation of each two-drug combination and by assessing whether combined inhibition of EGFR, SOS1, and SHP2 would be more effective than two drug combinations of these inhibitors. To assess each two-drug combination, H1975 cells cultured under 3D spheroid conditions were treated with dose-equivalent combinations of osimertinib, BAY-293, and RMC-4550, assessed for cell viability, and subjected to isobologram analysis to assess drug-drug synergy. Each two-drug combination showed synergy at three different DEQ ratios (*Figure 7C*), suggesting that inhibition of any two proximal signaling proteins may be an effective therapeutic regimen to treat EGFR-mutated cancer. To assess whether adding a third proximal inhibitor to each two-drug combination would further enhance synergistic inhibition of spheroid survival, each two-drug combination was mixed at 1:1 ratio, and then a third proximal pathway

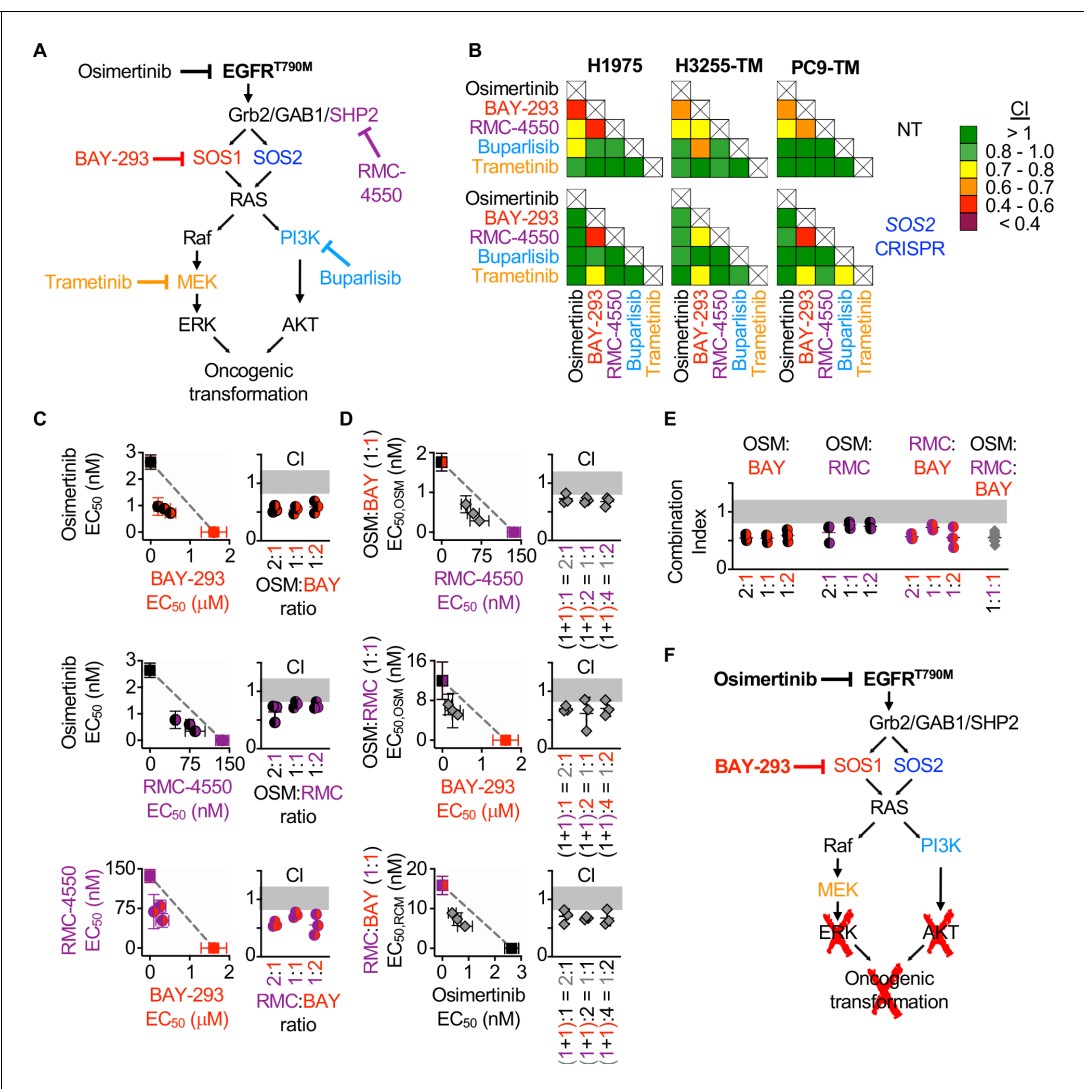

**Figure 7.** Assessment of the EGFR/RAS pathway 'inhibitor landscape' suggests that combination therapies inhibiting mutated EGFR, SOS1, and SHP2 have therapeutic potential in EGFR-mutated NSCLC. (**A**) Signaling diagram showing EGFR/RAS pathway inhibitors that were assessed for pairwise synergy by isobologram analysis using 50:50 dose-equivalent mixes of each drug pair. (**B**) Heat map of Combination Indices from isobologram analyses of the indicated drug-drug combinations in NT and *SOS2* KO NSCLC cell lines. Synergistic combinations are indicated in yellow, orange, or red. Data are presented as the mean from three independent experiments. (**C-D**) Isobologram analysis and Combination Index (CI) from dose-equivalent treatments of 3D spheroid cultured NCI-H1975 cells treated with the indicated two-drug (**C**) or three-drug (**D**) combinations of osimertinib (black), RMC-4550 (purple), and BAY-293 (red). For three drug combination, the two drugs indicated on the y-axis were held at a 1:1 ratio, and then mixed at dose equivalent ratiow with the third drug. CI values indicate enhanced synergy beyond the two drug combination on the y-axis of the isobologram plot and are calculated based on the y-axis drug combination calculated a s single drug treatment. Additive effects occur on the dashed lines of the isobologram plot and have a CI 0.8–1.2 (gray box), whereas synergistic interactions fall below the dashed lines and have a CI <0.8. (**E**) Combination indices from two-drug combinations of osimertinib (black), RMC-4550 (purple), and BAY-293 (red) mixed at 2:1, 1:1, or 1:2 ratios or the three drug combination at a 1:1:1 ratio (grey). CI are calculated based on three individual drug treatments. (**F**) Signaling model based on data from Figures 1–7 showing that combined targeting of mutated EGFR and SOS1 provides sufficient vertical inhibition of upstream signaling to inhibit RAS effector signaling and block oncogenic transformation. This synergistic inhibition can be further enhanced by SHP2 inhibition, providing multiple potential drug combinations for therapeutic intervention in EGFR-mutated NSCLC. Isobologram and CI data are presented as mean +/- s.d. from three independent experiments. For each experiment, three technical replicates were assessed.

The online version of this article includes the following source data for figure 7:

**Source data 1.** Assessment of the EGFR/RAS pathway 'inhibitor landscape' suggests that combination therapies inhibiting mutated EGFR, SOS1, and SHP2 have therapeutic potential in EGFR-mutated NSCLC.

inhibitor was added to give the indicated three-drug mixtures (*Figure 7D*). Isobologram analysis of these three drug mixtures revealed that addition of a third proximal pathway inhibitor to any two-drug combination of osimertinib, BAY-293, and RMC-4550 further enhanced synergy above what was observed for each two-drug combination (*Figure 7D*). Finally, comparing the combination index for the three-drug combination at a 1:1:1 ratio when each drug is treated independently versus the two-drug combinations showed marked synergy for the three drug combination, but that this synergy was not significantly enhanced compared to the combination of osimertinib and BAY-293 (*Figure 7E*). These data indicate that vertical inhibition of proximal EGFR signaling with the combination of osimertinib and a SOS1 inhibitor may be the most the most effective therapeutic combination to treat *EGFR*-mutated NSCLC.

## Discussion

Activating *EGFR* mutations are found in 10–30% of lung adenocarcinomas and are the major cause of lung cancer in never smokers. The third-generation EGFR-TKI osimertinib enhances both progression-free (*Soria et al., 2018*) and overall survival (*Ramalingam et al., 2020*) compared to first generation EGFR-TKIs and is now considered first-line treatment in EGFR-mutated NSCLC. Osimertinib resistance often develops via activation of parallel RTK pathways (*Mancini et al., 2018*; *Romaniello et al., 2018*; *La Monica et al., 2017*), and broad inhibition RTK signaling may enhance osimertinib efficacy and delay therapeutic resistance. Here, we demonstrate that inhibition of the common RTK signaling intermediate SOS1 using BAY-293 showed marked synergy with osimertinib in 3D spheroid-cultured EGFR-mutated NSCLC cells. Our observations that (i) osimertinib–BAY-293 synergy was only observed in 3D spheroids but not in adherent (2D) cultures and (ii) synergy between RTK-signaling intermediates and osimertinib was not broadly applicable to EGFR downstream signaling components but was limited to proteins upstream of RAS reveal novel insights into pharmacologic studies assessing therapeutics designed to treat NSCLC.

While most studies designed to identify or test therapeutic targets to treat cancer are done in 2D adherent culture, a growing body of evidence suggests that pharmacologic assessment of novel therapeutics must be performed in 3D culture systems (*Nunes et al., 2019*). Here, there are many different 3D model systems available that vary in both ease-of-use and complexity of the system. The simplest systems employ non-scaffold-dependent monoculture of cancer cell lines where spheroids are either generated using hanging-drop methodology, magnetic levitation, or using ultra-low attachment plates. More complex systems include embedding spheroids in an extracellular matrix (Matrigel, collagen, gelatin, or a synthetic hydrogel) either as a cancer cell line monoculture or in combination with cancer-derived fibroblasts, or using specialized microfluidics or culturing cancer-derived organoids. These methods are have been thoroughly reviewed elsewhere (*Langhans, 2018*). In the current study, we use ultra-low attachment plates of monoculture NSCLC cell lines as these have the advantage of recapitulating in vivo findings while allowing for dose-response studies done at scale (*Mittler et al., 2017*).

In NSCLC, multiple studies have now revealed the importance of 3D culture systems in order to recapitulate in vivo findings. *EGFR*-mutated cells show differential RTK expression and phosphorylation in 3D versus 2D conditions (*Ekert et al., 2014*) and respond more robustly to EGFR-TKIs in 3D cultures compared to 2D settings (*Figure 1* and *Jacobi et al., 2017*); KRAS-mutated cell lines deemed 'KRAS-independent' in 2D culture (*Balbin et al., 2013*; *Singh et al., 2009*; *Singh et al., 2012*; *Scholl et al., 2009*; *Lamba et al., 2014*) still require KRAS for anchorage-independent growth (*Fujita-Sato et al., 2015*; *Rotem et al., 2015*; *Zhang et al., 2006*; *McCormick, 2015*), and some KRAS$^{G12C}$-mutated NSCLC cell lines respond to KRAS(G12C) inhibitors in 3D culture and in vivo but not in 2D adherent culture (*Janes et al., 2018*). The relevance of 3D culture systems extends to the identification of novel therapeutic targets and therapeutic combinations. We recently showed that SOS2 is specifically required for PI3K-dependent protection from anoikis in KRAS-mutated NSCLC cells (*Sheffels et al., 2019*) and *SOS2* deletion synergizes with MEK inhibition to kill *KRAS* mutated cells only under 3D culture conditions (*Sheffels et al., 2018*). Here, we show marked synergy between vertical inhibition of EGFR and SOS1 in *EGFR* mutated cancer cells, but only under 3D culture conditions (*Figure 2*). CRISPR screens performed in spheroid cultures of KRAS- and EGFR-mutated NSCLC cell lines more accurately reproduce in vivo findings and identify drivers of oncogenic growth compared to screens performed in 2D cultures (*Han et al., 2020*). Intriguingly, in this

study SOS1 was essential for 3D spheroid survival but not 2D spheroid growth of both EGFR- and KRAS-mutated cells, and a recently accepted publication assessing a novel SOS1 inhibitor showed that it was more effective in 3D compared to 2D culture (*Hofmann et al., 2020*). These data are in complete agreement with our data from *Figure 1* showing the requirement for SOS1 in 3D transformation but not 2D proliferation, and support our conclusion that SOS1 is an important therapeutic target in *EGFR*-mutated NSCLC. We hypothesize the requirement for SOS1 (and SOS2) to promote oncogenic growth in 3D versus proliferation in 2D culture are due to the requirement for PI3K signaling to promote cell survival in 3D but not 2D. Downstream of EGFR activation, the threshold for Raf/MEK/ERK versus PI3K/AKT pathway activation are drastically different, so small amounts of EFGR signaling (in the presence of either SOS1 or SOS2) promote Raf/MEK/ERK signaling, whereas high levels of EGFR signaling are required to activate the PI3K/AKT pathway (*Fortian and Sorkin, 2014*). While this hypothesis remains to be tested, we speculate that depending on the specific oncogenic contexts, either SOS1 or SOS2 inhibition will be sufficient to modulate RTK signaling and change the threshold for PI3K signaling, thereby affecting oncogenic growth. These data suggest that future studies assessing novel therapeutics to treat lung adenocarcinomas must be performed in a 3D setting, and that SOS1 and SOS2 might be ubiquitous therapeutic targets in RTK-driven tumors.

Osimertinib resistant can occur via oncogenic shift to alternative RTKs including c-MET (*Shi et al., 2016*), HER2 and/or HER3 (*Mancini et al., 2018*; *Romaniello et al., 2018*; *La Monica et al., 2017*), IGF1R (*Park et al., 2016*), and AXL (*Kim et al., 2019*; *Taniguchi et al., 2019*; *Jimbo et al., 2019*; *Namba et al., 2019*). The variety of RTK bypass pathways that can lead to osimertinib resistance suggests that broad inhibition of RTK signaling may be a more effective therapeutic strategy than any individual RTK inhibitor to limit osimertinib resistance, whereas once resistance via oncogenic shift to an alternative RTK occurs then inhibition of the upregulated RTK would have therapeutic benefit. Toward this end, Phase I and II clinical trials are currently examining whether combining osimertinib with inhibitors of AXL (DS-1205c, NCT03255083) or c-MET (teponitib, NCT03940703; savolitinib, NCT03778229) are effective in patients who have progressed on osimertinib treatment.

Combining osimertinib with a MEK inhibitor can enhance osimertinib efficacy (*Eberlein et al., 2015*; *Tricker et al., 2015*; *Ichihara et al., 2017*; *Shi et al., 2017*; *Della Corte et al., 2018*) and Phase II clinical trials are currently underway to assess combining osimertinib with the MEK inhibitor selumetinib in EGFR-mutated NSCLC (NCT03392246), although resistance to combined osimertinib and MEK inhibition still occurs (*Tricker et al., 2015*). In a recent study designed to understand resistance to combined osimertinib and MEK inhibition, *Kurppa et al., 2020* show that combining osimertinib with the MEK inhibitor trametinib results in *EGFR*-mutated cells entering a senescent state that is dependent on the activation of the Hippo pathway effector YAP and its transcription-factor-binding partner TEAD (*Kurppa et al., 2020*). Inhibition of YAP/TEAD signaling overcame this senescence and enhanced killing of EGFR-mutated cells (*Kurppa et al., 2020*). EGFR-signaling drives YAP nuclear translocation and transcriptional regulation through PI3K-PDK1 signaling (*Fan et al., 2013*; *Xia et al., 2018*; *Tumaneng et al., 2012*). This suggest that therapeutic combinations able to synergistically inhibit both Raf/MEK/ERK and PI3K/AKT effector signaling should overcome YAP-dependent senescence and treat *EGFR*-mutated NSCLC.

Here, we show that osimertinib does not broadly synergize with inhibitors of downstream EGFR/RAS/RAS effector signaling. Instead, we found that synergy was limited to combinations of osimertinib with inhibitors of proximal EGFR signaling intermediates SOS1 and SHP2 (*Figure 7*). Further, SOS1 inhibition significantly enhanced osimertinib-dependent inhibition of both Raf/MEK/ERK and PI3K/AKT signaling (*Figure 6*), whereas inhibition of individual downstream Raf/MEK/ERK or PI3K/AKT effector pathways did not synergize with osimertinib (*Figure 7*) to inhibit 3D spheroid growth. We hypothesize that these two findings are inexorably linked, so that any potential therapeutic must synergize with osimertinib to inhibit all downstream RAS effector signaling to show drug-drug synergy in 3D culture. In support of this idea, previous studies showed inhibition of SRC family kinases (SFK) potentiated osimertinib to a much greater extent than either MEK or PI3K inhibition (*Ichihara et al., 2017*), and that SFK inhibition synergized with osimertinib to inhibit both Raf/MEK/ERK and PI3K/AKT signaling (*Ichihara et al., 2017*; *Watanabe et al., 2017*).

There remain several open questions regarding SOS1 inhibition as a therapeutic strategy to limit osimertinib resistance. First, does SOS1 inhibition enhance osimertinib efficacy in vivo using xenograft studies? While BAY-293 shows tremendous specificity toward SOS1 (*Figure 1—figure supplement 1 and 2*, and *Hillig et al., 2019*) and is a very useful tool compound for in vitro studies, it has

limited bioavailability making it unsuitable for in vivo use. Thus, new SOS1 inhibitors that can be used in vivo are needed to move SOS1 forward as a therapeutic target. Intriguingly, while this paper was under review Boehringer Ingelheim reported two orally available SOS1 inhibitors suitable for in vivo studies (*Hofmann et al., 2020*). They found that SOS1 inhibition could overcome MEK inhibitor resistance in *KRAS*-mutated cell lines and that the combination of SOS1 and MEK inhibition showed marked show efficacy in *KRAS*-mutated cell lines and xenograft models. They are now moving one of these compounds into Phase I safety trials for *KRAS* mutated solid tumors (BI-1701963, NCT04111458). It will be exciting to assess whether these new SOS1 inhibitors work in combination with osimertinib to limit the growth *EGFR*-mutated tumors. Further, these studies will be necessary to translate SOS1-targeted therapies for use in *EGFR*-mutated lung adenocarcinoma. Second, does SOS1 inhibition actually limit the development of osimertinib resistance? While outside the scope of the current paper, it will be intriguing to use in vitro models of EGFR-TKI resistance (*Tricker et al., 2015*) to assess whether SOS1 inhibition can block the development of osimertinib resistance. Third, while we have focused on the RAF/MEK/ERK and PI3K/AKT effector pathways as the major contributors to mutant *EGFR*-driven NSCLC, there are many different effector pathways downstream of RAS that may be SOS1-dependent and contribute to the oncogenic phenotype. Here, and unbiased approach at understanding the individual and combined effects of osimertinib and SOS1 inhibition on RAS activation (to validate relatively new SOS1 inhibitors) and RAS effector signaling would provide valuable insight into how these therapies alter EGFR-driven signaling in NSCLC.

Overall, our data suggest that inhibitors of proximal signaling may be the most efficacious therapeutics to combine with osimertinib to treat EGFR-mutated tumors. Toward this end, Phase I trials are currently underway assessing the combination of osimertinib and the SRC inhibitor dasatinib (NCT02954523) in *EGFR*-mutated NSCLC, and recently developed SOS1 (BI-1701963, NCT04111458) and SHP2 (JAB-3068, NCT03565003; RMC-4630, NCT03634982) inhibitors have entered Phase I safety trials. Our study provides a framework for the systematic, preclinical assessment of therapeutic combinations designed to treat EGFR-mutated cancer cells. We show both how to use basic pharmacologic principles to assess drug-drug synergy and that these combinations must be assessed under 3D culture conditions. Using this framework, we show that the combination of osimertinib and the SOS1 inhibitor BAY-293 shows marked efficacy in 3D spheroid culture and should be pursued as a therapeutic option to treat EGFR-mutated lung adenocarcinoma.

# Materials and methods

## Key resources table

| Reagent type (species) or resource | Designation | Source or reference | Identifiers | Additional information |
|---|---|---|---|---|
| Cell line (*Homo sapiens*) | Lung; adenocarcinoma; non-small cell lung cancer | Obtained from Udayan Guha, available at ATCC | NCI-H1975 CRL-5908 RRID:CVCL_UE30 | |
| Cell line (*Homo sapiens*) | Lung; adenocarcinoma; epithelial | Obtained from Udayan Guha, available at ATCC | HCC827 CRL-2868 RRID:CVCL_DH92 | |
| Cell line (*Homo sapiens*) | Lung; adenocarcinoma; non-small cell lung cancer | Obtained from Udayan Guha, available at NCI-DTP or ATCC | NCI-H3255 CRL-2882NCI-DTP Cat# NCI-H3255, RRID:CVCL_6831 | |
| Cell line (*Homo sapiens*) | Lung; adenocarcinoma; non-small cell lung cancer | *de Bruin et al., 2014* | NCI-H3255TM | |
| Cell line (*Homo sapiens*) | Dermal fibroblast (normal, Adult) | Obtained from Udayan Guha, available at Millipore Sigma or BCRJ | PC9 #90071810 BCRJ Cat# 0331, RRID:CVCL_B260 | |
| Cell line (*Homo sapiens*) | Lung; adenocarcinoma; non-small cell lung cancer | *Engelman et al., 2006* | PC9-TM | |

*Continued on next page*

*Continued*

| Reagent type (species) or resource | Designation | Source or reference | Identifiers | Additional information |
|---|---|---|---|---|
| Cell line (*Homo sapiens*) | Kidney; epithelial fibroblast (fetus) | ATCC | HEK-293T ATCC Cat# CRL-3216, RRID:CVCL_0063 | |
| Other | TransIT-Lenti | Mirus | Catalogue # MIR 6605 | Lentiviral transduction reagent |
| Other | MISSION Lentiviral packaging mix | Millipore Sigma | Catalogue # SHP001 | |
| Other | Bovine Serum Albumin | Millipore Sigma | Catalogue # A8022 | Cell culture reagent for ACL-4 media |
| Other | apo-Transferrin (human) | Millipore Sigma | Catalogue # T5391 | Cell culture reagent for ACL-4 media |
| Other | Sodium Selenite | Millipore Sigma | Catalogue # S9133 | Cell culture reagent for ACL-4 media |
| Other | Hydrocortisone | Millipore Sigma | Catalogue # H0135 | Cell culture reagent for ACL-4 media |
| Other | Ethanolamine | Millipore Sigma | Catalogue # E0135 | Cell culture reagent for ACL-4 media |
| Other | O-Phosphoryl ethanolamine | Millipore Sigma | Catalogue # P0503 | Cell culture reagent for ACL-4 media |
| Other | 3,3',5-Triiodo-L-thyronine [T3] | Millipore Sigma | Catalogue # T5516 | Cell culture reagent for ACL-4 media |
| Other | Sodium Pyruvate | Millipore Sigma | Catalogue # P4562 | Cell culture reagent for ACL-4 media |
| Other | HEPES | Invitrogen | Catalogue # 15630–080 | Cell culture reagent for ACL-4 media |
| Other | Epidermal Growth Factor [EGF] | Millipore Sigma | Catalogue # E4127 | Cell culture reagent for ACL-4 media |
| Other | Recombinant Human Insulin | Millipore Sigma | Catalogue # I9278 | Cell culture reagent for ACL-4 media |
| Other | AggreWell 400 low-attachment culture plates | Stem Cell | Catalogue # 34415 | |
| Other | ultra-low attachment 96-well round bottomed plates | Corning Corstar | Catalogue # 7007 | |
| Other | Nunc Nucleon Sphera microplates | ThermoFisher | Catalogue # 174929 | |
| Other | coated 96-well white-walled CulturePlates | Perken Elmer | Catalogue # 6005688 | |
| Antibody | anti-Sos 1 Antibody (C-23): sc-256, rabbit polyclonal | Santa Cruz | sc-256 | (1:500) |
| Antibody | anti-SOS2 antibody (C-19): sc-258, rabbit polyclonal | Santa Cruz | sc-258 | (1:500) |
| Antibody | anti-β-actin antibody AC15, mouse monoclonal | Millipore Sigma | #A1978 | (1:5000) |
| Antibody | anti-Phospho-EGF Receptor (Tyr1068) (D7A5) XP Rabbit mAb #3777 | Cell Signaling Technology | #3777 | (1:1000) |
| Antibody | anti-phospho p44/42 MAPK (Erk1/2) (Thr202/Tyr204) (D13.14.4E) XP Rabbit mAb #4370 | Cell Signaling Technology | #4370 | (1:1000) |

*Continued on next page*

*Continued*

| Reagent type (species) or resource | Designation | Source or reference | Identifiers | Additional information |
|---|---|---|---|---|
| Antibody | anti-p44/42 MAPK (Erk1/2) (L34F12) Mouse mAb #4696 | Cell Signaling Technology | #4696 | (1:1000) |
| Antibody | anti- Phospho-Akt (Ser473) (D9E) XP Rabbit mAb #4060 | Cell Signaling Technology | #4060 | (1:1000) |
| Antibody | anti- Akt (pan) (40D4) Mouse mAb #2920 | Cell Signaling Technology | #2920 | (1:1000) |
| Antibody | anti-HSP 90α/β Antibody (H-114): sc-7947 | Santa Cruz | #sc-7947 | (1:1000) |
| Antibody | anti-EGF Receptor (D38B1) XP Rabbit mAb #4267 | Cell Signaling Technology | #4267 | (1:1000) |
| Recombinant DNA Reagent | pLentiCrispr v2 | *Sanjana et al., 2014* | | |
| Other | CellTiter-Glo 2.0 | Promega | G9243 | |
| Recombinant DNA reagent | pLentiCrispr. NT | *Sheffels et al., 2018* | NT | sgRNA: CCATATCG GGGCGAGACATG |
| Recombinant DNA reagent | pLentiCrispr. SOS2-9 | *Sheffels et al., 2018* | SOS2-9 | sgRNA: GAGAACA GTCCGAAATGGCG |
| Recombinant DNA reagent | pLentiCrispr. SOS1-1 | This manuscript | SOS1-1 | sgRNA: GGGCAGC TGCTGCGCCTGCA |
| Recombinant DNA reagent | pLentiCrispr. SOS1-2 | This manuscript | SOS1-2 | sgRNA: GCATCCT TTCCAGTGTACTC |
| Recombinant DNA reagent | pLentiCrispr. SOS1-3 | This manuscript | SOS1-3 | sgRNA: TATTCTG CATTGCTAGCACC |
| Recombinant DNA reagent | pLentiCrispr. SOS1-4 | This manuscript | SOS1-4 | sgRNA: AGTGGCA TATAAGCAGACCT |
| Recombinant DNA reagent | pLentiCrispr. SOS1-5 | This manuscript | SOS1-5 | sgRNA: ATTGCAA GAGACAATGGACC |
| Recombinant DNA reagent | pLentiCrispr. SOS1-6 | This manuscript | SOS1-6 | sgRNA: GCTTATAT GCCACTCAACTG |
| Recombinant DNA reagent | pLentiCrispr. SOS1-7 | This manuscript | SOS1-7 | sgRNA: GAAGGAA CTCTTACACGTGT |
| Recombinant DNA reagent | pLentiCrispr. SOS1-8 | This manuscript | SOS1-8 | sgRNA: CTATTGG GTGTAAGGTGAGC |

## Cell culture

Cell lines were cultured at 37°C and 5% $CO_2$. HCC827, NCI-H1975, PC9, and PC9-TM cells were maintained in Roswell Park Memorial Institute medium (RPMI), each supplemented with 10% fetal bovine serum and 1% penicillin-streptomycin. Cell lines were authenticated by STR profiling and confirmed as mycoplasma negative. EGFR mutations were confirmed by Sanger sequencing. H3255 and H3255-TM were maintained in ACL4 medium formulated in DMEM:F-12 including: Bovine Serum Albumin 0.5% (w/v) (Sigma cat no. A8022), apo-Transferrin (human) (Sigma cat no. T5391) 0.01 mg/mL, Sodium Selenite (Sigma cat no. S9133) 25 nM, Hydrocortisone (Sigma cat no. H0135) 50 nM, Ethanolamine (Sigma cat no. E0135) 0.01 mM, O-Phosphorylethanolamine (Sigma cat no. P0503) 0.01 mM, 3,3',5-Triiodo-L-thyronine [T3] (Sigma cat no. T5516) 100pM, Sodium Pyruvate (Sigma cat no. P4562), HEPES (Invitrogen cat no 15630–080) 10 mM, Epidermal Growth Factor [EGF] 1 ng/mL, Recombinant Human Insulin (Sigma cat no. I9278) 0.02 mg/mL, and 1% penicillin-streptomycin. For signaling experiments, cells were seeded in 24-well micropatterned AggreWell 400 low-attachment culture plates (Stem Cell # 34415) at $1.2 \times 10^6$ cells/well in 2 mL of medium. 24 hr post-plating, half of the media was carefully replaced with fresh media to not disturb the spheroids. At 48 hr, 1 mL

media was removed and replaced with 2 x inhibitor. Cells were treated with inhibitor for 6 hr and then collected for cell lysis and western blot analysis.

## Cell lysis and western blot analysis

Cells were lysed in RIPA buffer (1% NP-40, 0.1% SDS, 0.1% Na-deoxycholate, 10% glycerol, 0.137 M NaCl, 20 mM Tris pH [8.0], protease (Biotool #B14002) and phosphatase (Biotool #B15002) inhibitor cocktails) for 20 min at 4°C and spun at 10,000 RPM for 10 min. Clarified lysates were boiled in SDS sample buffer containing 100 mM DTT for 10 min prior to western blotting. Proteins were resolved by sodium dodecyl sulfate-polyacrylamide (Criterion TGX precast) gel electrophoresis and transferred to nitrocellulose membranes. Western blots were developed by multiplex Western blotting using anti-SOS1 (Santa Cruz sc-256; 1:500), anti-SOS2 (Santa Cruz sc-258; 1:500), anti-β-actin (Sigma AC-15; 1:5,000), anti-pEGFR (Cell Signaling 3777; 1:1000), anti-EGFR (Cell Signaling 4267; 1:1000), anti-pERK1/2 (Cell Signaling 4370; 1:1,000), anti-ERK1/2 (Cell Signaling 4696; 1:1000), anti-pAKT Ser$^{473}$ (Cell Signaling 4060; 1:1000), anti-AKT (Cell Signaling 2920; 1:1000), anti-HSP90 (Santa Crux sc-7947, 1:1000), primary antibodies. Anti-mouse and anti-rabbit secondary antibodies conjugated to IRDye680 or IRDye800 (LI-COR; 1:10,000) were used to probe primary antibodies. Western blot protein bands were detected and quantified using the Odyssey system (LI-COR). For quantification of SOS1 and SOS2 abundance, samples were normalized to either β-actin or HSP90. For quantification of pERK and pAKT, samples were normalized to a weighted average of HSP90, β-actin, total ERK1/2, total AKT, and total EGFR (*Janes, 2015*).

## Proliferation studies

For 2D proliferation assays, $5 \times 10^2$ cells were seeded on cell culture-coated 96-well white-walled CulturePlates (Perkin Elmer #6005688). Cells were lysed with CellTiter-Glo 2.0 Reagent (Promega), and luminescence was read using a Bio-Tek Cytation five multi-mode plate reader. Cell number was assessed 24 hr after plating to account for any discrepancies in plating (Day 1), and then on days 3, 5, and 7. Data were analyzed as an increase in luminescence over Day 1.

## Transformation studies

H3255 and H3255-TM cells were seeded in 0.32% Nobel agar at $2 \times 10^4$ cells per 35 mm dish to assess anchorage-independent. Soft agar colonies were counted 28 days after seeding. For all other cell lines spheroid growth assessed in ultra-low attachment 96-well round bottomed plates (Corning Costar #7007), cells were seeded at 500 cells per well. Images were taken 24 hr after plating to assess initial spheroid size, and then 7, 14, and 21 days later to assess transformation. Cell number was assessed in parallel plates at 0, 7, 14, and 21 days using CellTiter-Glo 2.0 reagent.

## sgRNA studies

A non-targeting (NT) single guide RNA (sgRNA), a SOS2-targeted sgRNA (*Sheffels et al., 2018*), and eight potential SOS1-targeted sgRNAs previously used to target SOS1 in a genome-wide CRISPR screen (*Munoz et al., 2016*) were each cloned into pLentiCRISPRv2 as previously described (*Sanjana et al., 2014*). SOS1-2 was chosen as the SOS1 sgRNA for the study, and SOS2-9 was chosen as previously described (*Sheffels et al., 2018*). For studies in *Figure 1*, cells were infected lentivirus to express the given sgRNA with Cas9, and cells were selected for 10 days with puromycin prior to Western blotting. Cell lysates were probed for SOS1 or SOS2, and only cell populations showing grater that 80% SOS deletion within the overall population were used. Importantly, cell clones were not used, rather cell populations where > 80% of cells showed SOS deletion were used to minimize clonal effects. Independent infections were used for each experiment.

| Construct | sgRNA |
|---|---|
| NT | CCATATCGGGGCGAGACATG |
| SOS2-9 | GAGAACAGTCCGAAATGGCG |
| SOS1-1 | GGGCAGCTGCTGCGCCTGCA |

*Continued on next page*

*Continued*

| Construct | sgRNA |
|-----------|-------|
| SOS1-2 | GCATCCTTTCCAGTGTACTC |
| SOS1-3 | TATTCTGCATTGCTAGCACC |
| SOS1-4 | AGTGGCATATAAGCAGACCT |
| SOS1-5 | ATTGCAAGAGACAATGGACC |
| SOS1-6 | GCTTATATGCCACTCAACTG |
| SOS1-7 | GAAGGAACTCTTACACGTGT |
| SOS1-8 | CTATTGGGTGTAAGGTGAGC |

## Production of recombinant lentiviruses

Lentiviruses were produced by co-transfecting MISSION lentiviral packaging mix (Sigma) into 293 T cells using Mirus *Trans*IT-Lenti transfection reagent (Mirus Bio # MIR6605) in Opti-MEM (Thermo Scientific #31-985-062). At 48 hr post-transfection, viral supernatants were collected and filtered. Viral supernatants were then either stored at −80℃ or used immediately to infect cells in combination with polybrene at 8 μg/mL. 48 hr post-infection, cells were selected in 4 μg/mL Puromycin (Invitrogen). Twelve days after selection, cells were analyzed for SOS1 and SOS2 expression and plated for proliferation and transformation assays.

## Inhibitor studies

- 2D adherent studies – Cells were seeded at 500–1,000 cells per well in 100 μL in the inner-60 wells of 96-well white-walled culture plates (Perkin Elmer) and allowed to attach for 48 hr prior to drug treatment. Cells were treated with drug for 72 hr prior to assessment of cell viability using CellTiter-Glo 2.0.
- 3D adherent studies – Cells were seeded at 500–1,000 cells per well in 100 μL in the inner-60 wells of 96-well ultra-low attachment round bottomed plates (Corning #7007) or Nunc Nucleon Sphera microplates (ThermoFisher # 174929) and allowed to coalesce as spheroids for 48–72 hr prior to drug treatment. Cells were treated with drug for 96 hr prior to assessment of cell viability using CellTiter-Glo 2.0.

For all studies, outer wells (rows A and H, columns 1 and 12) were filled with 200 μL of PBS to buffer inner cells from temperature and humidity fluctuations. Triplicate wells of cells were then treated with increasing concentrations 100 μL of 2 × inhibitor at either a semilog (single drug dose response curves to determine $EC_{50}$) or a 1/3-log scale (isobologram and Bliss independence experiments) for 72 (adherent cultures) or 96 (spheroids) hr. Cell viability was assessed using CellTiter-Glo 2.0 (30 μL/well). Luminescence was assessed using a Bio-Tek Cytation five multi-mode plate reader. Data were normalized to the maximum luminescence reading of untreated cells, and individual drug $EC_{50}$ values were calculated using Prism eight by non-linear regression using log(inhibitor) vs. response with a variable slope (four parameters) to assess for differences in the Hill Coefficient between different drug treatments. For all drug-treatment studies, the untreated sample for each cell line was set to 100%. This would mask any differences in 3D cell proliferation seen between cell lines.

## Isobologram analysis

Dose equivalence was first determined by assessing individual-drug $EC_{50}$ values; individual-drug Hill Coefficients were determined to assure that the two drugs could be assessed for synergy by Lowe additivity. To generate dose-equivalent dose-response curves, the dose for each drug closest to the $EC_{50}$ on a 1/3-log scale was set as equivalent, and 10-point dose response curves were generated for each individual drug on either side of the equivalent dose to ensure the top (no drug effect) and bottom (maximal drug effect) were represented on the dose-response curve. 100 μL of drug each drug dose was added as outlined above. To generate dose-equivalent mixtures for isobologram analysis, equivalent doses of the two drugs were mixed at different ratios so that the total dose (100 μL) would be expected to have an equivalent effect on the cells if the two drugs were additive.

Drugs were mixed at either five (4:1, 2:1, 1:1, 1:2, and 1:4) or three (2:1, 1:1, and 1:2) different drug mixtures depending on the experiment. Cells were treated and $EC_{50}$ values for each individual drug or drug mixture based on each drug's dosing were determined for as outlined above. To generate an isobologram plot, the $EC_{50}$ of each individual drug was plotted as the x- or y-intercept, and the calculated contribution of each drug to the overall $EC_{50}$ for each DEQ mix is plotted as a single point $(EC_{50,A}, EC_{50,B})$ on the graph.

$$\text{Combination Index} = \frac{EC_{50}A_{mix}}{EC_{50}A_{alone}} + \frac{EC_{50}B_{mix}}{EC_{50}B_{alone}}$$

To calculate the combination index for each dose equivalent mixture, the calculated contribution of each drug to the overall $EC_{50}$ were used in the equation:

As an example, we will show data for one trial analyzing the combination of osimertinib and BAY-293 in 3D spheroid cultured H1975 cells in *Figure 2B*. The $EC_{50}$ values for each individual drug were first determined: −8.57 for osimertinib and −5.73 for BAY-293. Based on these $EC_{50}$ values, the dose equivalence was set at −8.67 for osimertinib −5.67 for BAY-293 (approximated $EC_{50}$ for each drug in bold), and the following 10-point dose response curves were generated:

| Osimertinib | −11 | −10.67 | −10.33 | −10 | −9.67 | −9.33 | -9 | −8.67 | −8.33 | -8 |
|---|---|---|---|---|---|---|---|---|---|---|
| BAY-293 | -8 | −7.67 | −7.33 | -7 | −6.67 | −6.33 | -6 | −5.67 | −5.33 | -5 |

Cells were then treated with the following volumes of each drug to generate seven dose-equivalent dose response curves:

| | | 4:1 mixture | 2:1 mixture | 1:1 mixture | 1:2 mixture | 1:4 mixture | |
|---|---|---|---|---|---|---|---|
| osimertinib | 100 µL | 80 µL | 66 µL | 50 µL | 34 µL | 20 µL | 0 µL |
| BAY-293 | 0 µL | 20 µL | 34 µL | 50 µL | 66 µL | 80 µL | 100 µL |

$EC_{50}$ values for each dose-response curve were then determined based on each drug's dosing:

| | OSM alone | 4:1 mixture | 2:1 mixture | 1:1 mixture | 1:2 mixture | 1:4 mixture | BAY alone |
|---|---|---|---|---|---|---|---|
| osimertinib $EC_{50}$ (nM) | 2.62 | 0.84 | 0.70 | 0.92 | 1.49 | 1.19 | 2.40 |
| BAY-293 $EC_{50}$ (µM) | 2.14 | 1.01 | 0.83 | 1.09 | 1.49 | 1.04 | 1.82 |

$EC_{50}$ values were then adjusted based on the amount of each drug that was put in the mixture to determine the contribution of each drug in the mixture to the overall $EC_{50}$. For example, the 4:1 mixture was 80% osimertinib, so the osimertinib EC50 for that mixture is multiplied by 0.8. The corresponding corrected $EC_{50}$ values and combination indices were:

| | OSM alone | 4:1 mixture | 2:1 mixture | 1:1 mixture | 1:2 mixture | 1:4 mixture | BAY alone |
|---|---|---|---|---|---|---|---|
| osimertinib $EC_{50}$ (nM) | 2.62 | 0.67 | 0.45 | 0.46 | 0.52 | 0.24 | 0 |
| BAY-293 $EC_{50}$ (µM) | 0 | 0.20 | 0.29 | 0.54 | 0.97 | 0.84 | 1.82 |
| Combination Index | | 0.40 | 0.34 | 0.44 | 0.65 | 0.46 | |

## Bliss independence analysis

Unlike Isobologram analysis, individual drug doses are not reduced for drug-drug combinations when performing Bliss independence analysis. For data in *Figure 2*, wells were treated with a full dose of each individual drug or drug combination in a 10 × 10 matrix of dose combinations for osimertinib and BAY-293 on a 1/3-log scale. Data were normalized to the maximum luminescence reading of untreated cells, and a heat-map depicting cell viability was generated using Prism 8. The Bliss index was calculated by first converting viability (on a scale of 0 to 1) for each treatment to the effect of each drug or drug combination, where 0 represents no effect and 1 represents 100% effect (no viable cells).

$$\text{effect} = 1 - \text{viability}$$

From the effect data, the expected effect for each drug combination is calculated:

$$\text{Expected effect} = \text{E}_\text{A} + \text{E}_\text{B} * (1 - \text{E}_\text{A})$$

$$\text{Expected effect} = \text{E}_\text{A} + \text{E}_\text{B} - \text{E}_\text{A} * \text{E}_\text{B}$$

The <u>Bliss Index</u> is the ratio of the expected effect/actual effect:

$$\text{Bliss Index} = (\text{expected effect}) / (\text{actual effect})$$

$$\text{Bliss Index} = (\text{E}_\text{A} + \text{E}_\text{B} - -\text{E}_\text{A} * \text{E}_\text{B}) / (\text{E}_\text{A+BMIX})$$

A Bliss Index of 1 indicates that the actual and expected effects are equivalent, and the effects of the two drugs are additive. Bliss Index < 1 indicates increasing synergy, whereas Bliss Index > 1 indicates antagonism.

<u>Excess over Bliss</u> is calculated by determining how much greater the actual effect of the drug combination is versus the expected effect, and is calculated as:

$$\text{Excess over Bliss} = 100^* [\text{actual effect} - -\text{expected effect}]$$

$$\text{Excess over Bliss} = 100^* [\text{E}_\text{A+BMIX} - (\text{E}_\text{A} + \text{E}_\text{B} - \text{E}_\text{A} * \text{E}_\text{B})]$$

An excess over Bliss of 0 indicates that the actual and expected effects are equivalent, and the effects of the two drugs are additive; values > 0 indicate increasing synergy, whereas values < 0 indicate antagonism.

Since synergy occurred at drug combinations at or just below the $EC_{50}$ values for each individual drug, Bliss experiments in *Figures 4* and *5*, drug mixtures were limited to 3 × 10 drug mixtures based on dose equivalence with mixtures at approximately 2:1, 1:1, and 1:2 mixes of the two drugs based on dose equivalence. Here, the doses used for one drug were held constant, and the second drug dose wash shifted by 1/3 log up or down to generate 2:1 and 1:2 mixtures. For example, for the combination of osimertinib and BAY-293 in H1975 cells, the following drug doses were used:

| Osimertinib (1:2 ratio of OSM:BAY) | −11.33 | −11 | −10.67 | −10.33 | −10 | −9.67 | −9.33 | -9 | −8.67 | −8.33 |
|---|---|---|---|---|---|---|---|---|---|---|
| Osimertinib (1:1 ratio of OSM:BAY) | −11 | −10.67 | −10.33 | −10 | −9.67 | −9.33 | -9 | −8.67 | −8.33 | -8 |
| Osimertinib (2:1 ratio of OSM:BAY) | −10.67 | −10.33 | −10 | −9.67 | −9.33 | -9 | −8.67 | −8.33 | -8 | −7.67 |
| BAY-293 (constant) | -8 | −7.67 | −7.33 | -7 | −6.67 | −6.33 | -6 | −5.67 | −5.33 | -5 |

## Three-drug isobologram analysis

For three-drug isobologram studies with osimertinib (EC$_{50}$ = −8.57), BAY-293 (EC$_{50}$ = −5.74), and RCM-4550 (EC$_{50}$ = −6.84), drugs were again mixed based on dose equivalency. The dose-equivalent 10-point dose-response curves for these drugs in 3D cultured H1975 cells were (approximated EC$_{50}$ for each drug in bold):

| Osimertinib | −11 | −10.67 | −10.33 | −10 | −9.67 | −9.33 | **-9** | −8.67 | −8.33 | **-8** |
|---|---|---|---|---|---|---|---|---|---|---|
| BAY-293 | -8 | −7.67 | −7.33 | -7 | −6.67 | −6.33 | -6 | **−5.67** | −5.33 | -5 |
| RMC-4550 | -9 | **−8.67** | −8.33 | -8 | −7.67 | −7.33 | -7 | **−6.67** | −6.33 | -6 |

Each two-drug combination was set as a single 'drug mixture' at a 1:1 ratio, and the third drug was combined with this drug mixture at 2:1, 1:1, and 1:2 drug ratios. To generate the proper two and three-drug mixtures for analysis, 21 total dose response curves were generated. The five dose-response curves on the right represent the mixtures used to generate the isobologram plots in *Figure 7D*. The other two two-drug mixtures in **bold** (two-drug 2:1 and 1:2 mixtures) were used to generate the isobologram plots in *Figure 7C*Combination indices were calculated based on whether addition of the third drug to each two-drug 1:1 mixture further enhanced synergy when added to the two-drug mixture.

[osimertinib:BAY-293] mixture vs. RCM-4550:

| | OSM:BAY 2:1 | OSM:BAY 1:2 | Osm:BAY 1:1 | (1+1):1 2:1 mixture | (1+1):2 1:1 mixture | (1+1):4 1:2 mixture | RCM alone |
|---|---|---|---|---|---|---|---|
| osimertinib | 66 µL | 34 µL | 50 µL | 33 µL | 25 µL | 17 µL | 0 µL |
| BAY-293 | 34 µL | 66 µL | 50 µL | 33 µL | 25 µL | 17 µL | 0 µL |
| RMC-4550 | 0 µL | 0 µL | 0 µL | 34 µL | 50 µL | 66 µL | 100 µL |

$$\text{Combination Index} = \frac{\text{EC}_{50}\,\text{OSM} + \text{BAY}_{3-\text{drug mix}}}{\text{EC}_{50}\,\text{OSM} + \text{BAY}_{50:50}} + \frac{\text{EC}_{50}\text{RCM}_{3-\text{drug mix}}}{\text{EC}_{50}\text{RCM}_{\text{alone}}}$$

[osimertinib:RCM-4550] mixture vs. BAY-293:

| | OSM:RCM 2:1 | OSM:RCM 1:2 | Osm:RCM 1:1 | (1+1):1 2:1 mixture | (1+1):2 1:1 mixture | (1+1):4 1:2 mixture | RCM alone |
|---|---|---|---|---|---|---|---|
| osimertinib | 66 µL | 34 µL | 50 µL | 33 µL | 25 µL | 17 µL | 0 µL |
| BAY-293 | 0 µL | 0 µL | 0 µL | 34 µL | 50 µL | 66 µL | 100 µL |
| RMC-4550 | 34 µL | 66 µL | 50 µL | 33 µL | 25 µL | 17 µL | 0 µL |

$$\text{Combination Index} = \frac{\text{EC}_{50}\,\text{OSM} + \text{RCM}_{3-\text{drug mix}}}{\text{EC}_{50}\,\text{OSM} + \text{RCM}_{50:50}} + \frac{\text{EC}_{50}\text{BAY}_{3-\text{drug mix}}}{\text{EC}_{50}\text{BAY}_{\text{alone}}}$$

[BAY-293:RCM-4550] mixture vs. osimertinib:

| | BAY:RCM 2:1 | BAY:RCM 1:2 | Bay:RCM 1:1 | (1+1):1 2:1 mixture | (1+1):2 1:1 mixture | (1+1):4 1:2 mixture | RCM alone |
|---|---|---|---|---|---|---|---|
| osimertinib | 0 µL | 0 µL | 0 µL | 34 µL | 50 µL | 66 µL | 100 µL |
| BAY-293 | 66 µL | 34 µL | 50 µL | 33 µL | 25 µL | 17 µL | 0 µL |
| RMC-4550 | 34 µL | 66 µL | 50 µL | 33 µL | 25 µL | 17 µL | 0 µL |

$$\text{Combination Index} = \frac{\text{EC}_{50}\,\text{BAY} + \text{RCM}_{3-\text{drug mix}}}{\text{EC}_{50}\,\text{BAY} + \text{RCM}_{50:50}} + \frac{\text{EC}_{50}\,\text{OSM}_{3-\text{drug mix}}}{\text{EC}_{50}\,\text{OSM}_{\text{alone}}}$$

To calculate the three-drug combination index where each drug was considered independently (*Figure 7E*), the following equation was used:

$$\text{Combination Index} = \frac{\text{EC}_{50}\,\text{OSM}_{3-\text{drug mix}}}{\text{EC}_{50}\,\text{OSM}_{50:50}} + \frac{\text{EC}_{50}\,\text{BAY}_{3-\text{drug mix}}}{\text{EC}_{50}\,\text{BAY}_{\text{alone}}} + \frac{\text{EC}_{50}\,\text{RCM}_{3-\text{drug mix}}}{\text{EC}_{50}\,\text{RCM}_{\text{alone}}}$$

## Acknowledgements

We thank Udayan Guha for NCI-H1975, HCC827, PC9, H3255 and H3255-TM cells and for helpful discussions throughout the project. We thank Julian Downward for PC9-TM cells. Funding: This work was supported by grants from the Congressionally Directed Medical Research Program to RLK (LC160222 and LC180213).

## Additional information

### Funding

| Funder | Grant reference number | Author |
|---|---|---|
| Congressionally Directed Medical Research Programs | LC160222 | Robert L Kortum |
| Congressionally Directed Medical Research Programs | LC180213 | Robert L Kortum |

The funders had no role in study design, data collection and interpretation, or the decision to submit the work for publication.

### Author contributions

Patricia L Theard, Robert L Kortum, Conceptualization, Resources, Data curation, Formal analysis, Supervision, Funding acquisition, Investigation, Methodology, Writing - original draft, Project administration, Writing - review and editing; Erin Sheffels, Conceptualization, Data curation, Formal analysis, Investigation, Writing - original draft, Writing - review and editing; Nancy E Sealover, Conceptualization, Investigation, Writing - review and editing; Amanda J Linke, David J Pratico, Investigation, Writing - review and editing

### Author ORCIDs

Robert L Kortum ⬥ https://orcid.org/0000-0002-1634-4882

### Decision letter and Author response

Decision letter https://doi.org/10.7554/eLife.58204.sa1
Author response https://doi.org/10.7554/eLife.58204.sa2

## Additional files

### Supplementary files

- Source data 1. Supplemental raw data.
- Transparent reporting form

### Data availability

All data generated or analysed during this study are included in the manuscript and supporting files. Source data files have been provided for Figures 1, 2, 3, 4, 5, 6, 7, and Figure 1—figure supplement 2 and Figure 3—figure supplement 1.

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
