## [Decision Letter]

**Acceptance summary:**

One of the weaknesses of cell culture studies to screen for anti-neoplastic drugs is that they lack the three-dimensionality or architectural complexity of tumors. The novel spheroid-based studies on non-small cell lung cancer in this paper advance the design of molecular targeted therapies by combining EGF Receptor kinase inhibitors with compounds that abrogate SOS and SHP2 signaling.

**Decision letter after peer review:**

Thank you for submitting your article "Marked Synergy by Vertical Inhibition of EGFR signaling in NSCLC: SOS1 as a therapeutic target in EGFR-mutated cancer" for consideration by *eLife*. Your article has been reviewed by four peer reviewers, and the evaluation has been overseen by a Reviewing Editor and Jonathan Cooper as the Senior Editor. The following individual involved in review of your submission has agreed to reveal their identity: Jeroen P Roose (Reviewer #4).

The reviewers have discussed the reviews with one another and the Reviewing Editor has drafted this decision to help you prepare a revised submission.

Summary:

Resistance to first and second generation EGFR TKI's is a clinically important problem. 3D cultures have the advantage of more closely resembling in vivo biology without the need for time-consuming and resource intensive in vivo studies and therefore exploring therapeutic synergies in 3D culture conditions is a worthwhile objective. Overall your work is meritorious, the findings are interesting, results are presented in a convincing manner, and the conclusions are generally plausible.

Essential revisions:

While the reviewers suggested that your findings would be even stronger were in vivo data included to corroborate the 3D cultures and comparison made with 2D culture data so to demonstrate that signaling changes are 3D specific. Please address the comments below and incorporate them into the Discussion section.

Similarly, exploring the therapeutic potential of SOS1/2 represents a promising novel pathway for targeting EGFR-TKI resistance. The work is original and interesting. The study methodology is provided in sufficient detail to assess the scientific rigor of the experiments and subsequent analysis. The authors are to be commended on very well written manuscript without distracting typographical errors; the figures are appealing, sequenced appropriately, clearly explained, and helpful schematics are included (e.g. Figure 2A-D, Figure 7A).

1) The authors contend 3D culture is more representative of in vivo with respect to therapeutic responses (and provide references for this). However, as this contention (superiority of 3D culture) is a central pillar of their work, the manuscript would be strengthened if the mechanisms that drive selective sensitivity to SOS1 deletion in 3D cultures (Figure 1) and synergy with combination therapies (Figures 2 and 3) were more thoroughly explored or discussed.

2) No in vivo data to corroborate the more representative nature of the 3D cultures is provided. The inclusion of in vivo experimental data would address questions related to the possibility of artifact related to SOS knockout in 3D culture conditions and similarly as to whether synergy will persist with BAY-293 treatment in vivo.

3) SOS1 deletion 'completely inhibited spheroid growth in both H1975 cells' (Figure 1C). As such it is unclear how this model is subsequently used to established specificity of BAY-293 for SOS1 in 3D Cx models. The latter data seem to conflict with the former in that 3D growth under SOS1 KO conditions is relatively unaffected (Figure 1F, Figure 1—figure supplement 2).

4) Combined EGFR and SOS1 inhibition inhibited Raf/MEK/ERK and PI3K/AKT signaling in 3D culture, suggesting a mechanism for the observed growth effects (Figure 6). Unclear if these downstream signaling changes are unique to 3D culture conditions (no 2D data); possibly a missed opportunity to provide an interesting mechanism for selective sensitization in 3D conditions.

The authors found that knockout of SOS1 or treatment with SOS1 inhibitor reduced EGFR-mutated lung cancer cell proliferation in 3D spheroid culture but not in 2D culture. Then, they showed that osimertinib had synergistic effects on inhibition of the EGFR mutated lung cancer cell proliferation with SOS1 inhibitor. They used several cancer cell lines carrying gefitinib-sensitive mutations with or without resistance mutations. Then they examined whether synergistic effects are observed among several compounds including inhibitors for Shp2, MEK or PI3K. They found that the combination of osimertinib and SOS1 inhibitor or osimertinib and Shp2 inhibitor had synergistic effects.

5) The experiments are well-done. It is reasonable to use two methods (combination index and Bliss index) for calculating the synergistic effects to confirm the reproducibility of the results. They clearly showed the synergistic effects between osimertinib and SOS1 inhibitor or osimertinib and Shp2 inhibitor. However, all the experiments were done by using in vitro 3D spheroid culture model. It would be very important to confirm their results in vivo, if synergistic effects of these combinations of the drugs are observed in preclinical models by using mouse xenograft.

6) It is unclear why SOS1 inhibitor had effects in 3D culture but not in 2D culture. Please show findings related to the reasonable mechanisms or discuss the potential mechanisms at least why they obtained such results.

7) They used one kind of targeting sequences for SOS1 and SOS2. In order to exclude the possibility of off-target effects, please perform single-cell cloning to have several clones or generate several knockout cells by using at least 2 kinds of targeting sequences. If you have these data please add. If not, please be clear in the paper.

8) Do disparate 3D spheroid (and organoid) culture systems exist? If so, please briefly summarize the 3D spheroid methodology and rationale, including pros and cons relative to other 3D culture techniques, in the Results section. Alternatively, if one uniform 3D culture technique is consistently used for human NSCLC cell line studies, please explain in detail.

9) In the Discussion, please address limitations of the current study and future directions. For example, SOS1 inhibitor, BAY-293 is not suitable for in vivo studies, and the authors should be transparent about that limitation and suggest future directions accordingly. In the absence of animal studies, it is important to describe whether in vivo model studies are necessary for translational and clinical development of the proposed treatment strategy, including discussion within the context of the discussed clinical trials (Discussion, last paragraph).

10) The study lacks direct evidence that osimertinib and BAY-293 dual therapy prevents EGFR TKI-refractory lung cancer growth. For example, one could envision a study where osimertinib-sensitive cells are continuously treated with osimertinib +/- BAY-293 over time (similar to how EGFR TKI resistant cells are made) to determine whether the addition of BAY-293 delays or prevents osimertinib resistance. Please address.

The reported synergy between EGFR inhibition and BAY-293 (but also Shp2 inhibition and BAY-293) and the lack of synergy between EGFR inhibition and targeting of RAF-MEK-ERK or PI3K-AKT is very intriguing and begs the question whether other effector pathways downstream of Ras are engaged (Discussion, fifth paragraph).

11) What is lacking here is a RasGTP pull down with BAY-293. Even only one cell line with the nice combo of 2D versus 3D and WT versus SOS1 CRISPR would be very valuable. The drug is still relatively new (2019, PNAS) and some validation would be important.

---

## [Author Response]

Essential revisions:While the reviewers suggested that your findings would be even stronger were in vivo data included to corroborate the 3D cultures and comparison made with 2D culture data so to demonstrate that signaling changes are 3D specific. Please address the comments below and incorporate them into the Discussion section.Similarly, exploring the therapeutic potential of SOS1/2 represents a promising novel pathway for targeting EGFR-TKI resistance. The work is original and interesting. The study methodology is provided in sufficient detail to assess the scientific rigor of the experiments and subsequent analysis. The authors are to be commended on very well written manuscript without distracting typographical errors; the figures are appealing, sequenced appropriately, clearly explained, and helpful schematics are included (e.g. Figure 2A-D, Figure 7A).1) The authors contend 3D culture is more representative of in vivo with respect to therapeutic responses (and provide references for this). However, as this contention (superiority of 3D culture) is a central pillar of their work, the manuscript would be strengthened if the mechanisms that drive selective sensitivity to SOS1 deletion in 3D cultures (Figure 1) and synergy with combination therapies (Figures 2 and 3) were more thoroughly explored or discussed.

A provisionally accepted publication from Boehringer Ingelheim at Cancer Discovery using their novel SOS1 inhibitor has corroborated the effect of SOS1 inhibition in 3D versus 2D culture (referenced). We have added text to the Discussion hypothesizing on the mechanism by which SOS1 inhibition or deletion is specifically important for 3D growth.

2) No in vivo data to corroborate the more representative nature of the 3D cultures is provided. The inclusion of in vivo experimental data would address questions related to the possibility of artifact related to SOS knockout in 3D culture conditions and similarly as to whether synergy will persist with BAY-293 treatment in vivo.

We appreciate the comment and agree that the next step is inhibiting SOS1 in vivo. Unfortunately, the tool compound BAY-293 is not suitable for in vivo studies due to bioavailability issues (Hillig et al., 2019). At the April AACR meeting, Marco Hoffman from Boehringer Ingelheim showed nice data on their orally available SOS1 inhibitors BI-3406 and BI-1701963, and a manuscript with these findings has been provisionally accepted at Cancer Discovery. We reference this manuscript, and look forward to testing these inhibitors and further using them inhibitors for in vivo studies once they are available.

We have added text to the Discussion to help clarify that BAY-293 could not be used for in vivo studies, and have referenced a recently accepted publication from Boehringer Ingelheim in Cancer Discovery (see below) describing their novel SOS1 inhibitors that are bioavailable in vivo.

3) SOS1 deletion 'completely inhibited spheroid growth in both H1975 cells' (Figure 1C). As such it is unclear how this model is subsequently used to established specificity of BAY-293 for SOS1 in 3D Cx models. The latter data seem to conflict with the former in that 3D growth under SOS1 KO conditions is relatively unaffected (Figure 1F, Figure 1—figure supplement 2).

We appreciate the comment and understand the confusion that our data normalization in Figure 1F and Figure 1—figure supplement 2 has caused. For Figure 1C, it is true that SOS1 deletion inhibited oncogenic transformation. In Figure 1F and Figure 1—figure supplement 2, we set the untreated sample for each individual cell line to 100%, and have reported the dose response curves relative to the untreated sample for each condition (NT vs. SOS1 KO vs. SOS2 KO). We have clarified this in the text, legends, and Materials and methods. Further, in both 1975 and 3255 cells, we do not see drastic 3D growth differences at 4 days in NT vs. SOS1 KO cells, so differences in the ‘baseline’ would not be expected.

4) Combined EGFR and SOS1 inhibition inhibited Raf/MEK/ERK and PI3K/AKT signaling in 3D culture, suggesting a mechanism for the observed growth effects (Figure 6). Unclear if these downstream signaling changes are unique to 3D culture conditions (no 2D data); possibly a missed opportunity to provide an interesting mechanism for selective sensitization in 3D conditions.

We appreciate the reviewer comments, and agree that these studies would be interesting. Unfortunately, we focused our studies early on to 3D culture, and do not have equivalent 2D signaling data to add to the current study. We like this premise, and will investigate the idea in future experiments. We hypothesize that the difference are not due to intrinsic differences to signaling responses to SOS1 inhibition in 2D vs. 3D, but rather to the cell’s need for PI3K signaling for protection from anoikis in 3D culture.

The authors found that knockout of SOS1 or treatment with SOS1 inhibitor reduced EGFR-mutated lung cancer cell proliferation in 3D spheroid culture but not in 2D culture. Then, they showed that osimertinib had synergistic effects on inhibition of the EGFR mutated lung cancer cell proliferation with SOS1 inhibitor. They used several cancer cell lines carrying gefitinib-sensitive mutations with or without resistance mutations. Then they examined whether synergistic effects are observed among several compounds including inhibitors for Shp2, MEK or PI3K. They found that the combination of osimertinib and SOS1 inhibitor or osimertinib and Shp2 inhibitor had synergistic effects.

We thank the reviewer for the kind comment.

5) The experiments are well-done. It is reasonable to use two methods (combination index and Bliss index) for calculating the synergistic effects to confirm the reproducibility of the results. They clearly showed the synergistic effects between osimertinib and SOS1 inhibitor or osimertinib and Shp2 inhibitor. However, all the experiments were done by using in vitro 3D spheroid culture model. It would be very important to confirm their results in vivo, if synergistic effects of these combinations of the drugs are observed in preclinical models by using mouse xenograft.

We appreciate the comment and agree that the next step is inhibiting SOS1 in vivo. Unfortunately, the tool compound BAY-293 is not suitable for in vivo studies due to bioavailability issues (Hillig et al., 2019). At the April AACR meeting, Marco Hoffman from Boehringer Ingelheim showed nice data on their orally available SOS1 inhibitors BI-3406 and BI-1701963, and this manuscript has been provisionally accepted at Cancer Discovery (they are submitting second revisions next week). We look forward to testing these inhibitors and further using them inhibitors for in vivo studies once they are published.

6) It is unclear why SOS1 inhibitor had effects in 3D culture but not in 2D culture. Please show findings related to the reasonable mechanisms or discuss the potential mechanisms at least why they obtained such results.

We have added text to the Discussion discussing potential mechanistic differences in 3D versus 2D cultures that cause SOS1 to only synergize with EGFR-TKIs in 3D.

7) They used one kind of targeting sequences for SOS1 and SOS2. In order to exclude the possibility of off-target effects, please perform single-cell cloning to have several clones or generate several knockout cells by using at least 2 kinds of targeting sequences. If you have these data please add. If not, please be clear in the paper.

We appreciate the comment and have updated the Materials and methods and the figure legend to clarify our findings. Indeed, we did use only one targeting vector for each, however, all of our experiments are done in pooled cell populations and not in ‘cell clones’. Whenever possible, we try to avoid using clones in the lab as they can show high variability in responsiveness that is not due to the biology being studied (in this case, SOS1 or SOS2). Instead, having a population of cells gives much more representative data. Also, each experiment where CRISPR was used to delete *SOS1* or *SOS2* was performed at least three independent times (three different infections with CRISPR virus on different dates). We have now made this clear in the manuscript.

8) Do disparate 3D spheroid (and organoid) culture systems exist? If so, please briefly summarize the 3D spheroid methodology and rationale, including pros and cons relative to other 3D culture techniques, in the Results section. Alternatively, if one uniform 3D culture technique is consistently used for human NSCLC cell line studies, please explain in detail.

We appreciate the comment. Indeed, although the method of spheroid culture we use is the most common (and easiest), there are several different methods of spheroid cultures. We have summarized these in the Discussion.

9) In the Discussion, please address limitations of the current study and future directions. For example, SOS1 inhibitor, BAY-293 is not suitable for in vivo studies, and the authors should be transparent about that limitation and suggest future directions accordingly. In the absence of animal studies, it is important to describe whether in vivo model studies are necessary for translational and clinical development of the proposed treatment strategy, including discussion within the context of the discussed clinical trials (Discussion, last paragraph).

We thank the reviewer for this comment and agree with the premise. This comment is in line with comments # 2 and #5 asking for in vivo studies. We have expanded our discussion of the limitations of BAY-293 for in situ studies only, and have expanded our discussion regarding the new Boehringer Ingelheim inhibitors provisionally accepted at Cancer Discovery.

10) The study lacks direct evidence that osimertinib and BAY-293 dual therapy prevents EGFR TKI-refractory lung cancer growth. For example, one could envision a study where osimertinib-sensitive cells are continuously treated with osimertinib +/- BAY-293 over time (similar to how EGFR TKI resistant cells are made) to determine whether the addition of BAY-293 delays or prevents osimertinib resistance. Please address.

We thank the reviewer for the comment, and agree that resistance studies (even in situ) are great experiments to do. We have added a discussion regarding this (and animal studies) future directions in the Discussion.

The reported synergy between EGFR inhibition and BAY-293 (but also Shp2 inhibition and BAY-293) and the lack of synergy between EGFR inhibition and targeting of RAF-MEK-ERK or PI3K-AKT is very intriguing and begs the question whether other effector pathways downstream of Ras are engaged (Discussion, fifth paragraph).

We thank the reviewer for the comment, and have added a discussion about other effector pathways.

11) What is lacking here is a RasGTP pull down with BAY-293. Even only one cell line with the nice combo of 2D versus 3D and WT versus SOS1 CRISPR would be very valuable. The drug is still relatively new (2019, PNAS) and some validation would be important.

We thank the reviewer for the comment, and agree that RAS pulldowns would have rounded out the signaling in Figure 6 nicely. These take time to set up / make CRISPR lines / etc. We have added a discussion of RAS pulldowns and further validation along with the need for resistance and animal studies.